# Finite-Time Global Optimality Convergence in Deep Neural Actor-Critic Methods for Decentralized Multi-Agent Reinforcement Learning

Zhiyao Zhang [* 1]   Myeung Suk Oh [* 1]   FNU Hairi [2]   Ziyue Luo [1]   Alvaro Velasquez [3]   Jia Liu [1]

## Abstract

Actor-critic methods for decentralized multi-agent reinforcement learning (MARL) facilitate collaborative optimal decision making without centralized coordination, thus enabling a wide range of applications in practice. To date, however, most theoretical convergence studies for existing actor-critic decentralized MARL methods are limited to the guarantee of a stationary solution under the linear function approximation. This leaves a significant gap between the highly successful use of deep neural actor-critic for decentralized MARL in practice and the current theoretical understanding. To bridge this gap, in this paper, we make the first attempt to develop a deep neural actor-critic method for decentralized MARL, where both the actor and critic components are inherently non-linear. We show that our proposed method enjoys a global optimality guarantee with a finite-time convergence rate of $\mathcal{O}(1/T)$, where $T$ is the total iteration times. This marks the first global convergence result for deep neural actor-critic methods in the MARL literature. We also conduct extensive numerical experiments, which verify our theoretical results.

## 1. Introduction

**1) Background and Motivations:** Decentralized Multi-agent reinforcement learning (MARL) (Littman, 1994; Lauer and Riedmiller, 2000; Lowe et al., 2017; Zhang et al., 2018; Omidshafiei et al., 2017), a generalization of traditional single-agent reinforcement learning (RL) (Kaelbling et al., 1996; Sutton, 2018; Arulkumaran et al., 2017) has found a wide range of applications in recent years, such as network recourse allocation (Cui et al., 2019; Liu et al., 2020b), autonomous driving (Sallab et al., 2017; Kiran et al., 2021; Yu et al., 2019), wireless random access network optimization (Luong et al., 2019; Nasir and Guo, 2019; Oh et al., 2025). Generally speaking, in decentralized MARL, multiple agents collaboratively learn an optimal policy to maximize long-term accumulative global rewards through local optimization and information sharing.

Among a large number of methods for solving MARL problems, the family of actor-critic methods have become increasingly popular. This is primarily due to the fact that actor-critic methods combine the strengths of two most fundamental and basic RL approaches, namely value-based and policy-based RL approaches, thus achieving the best of both worlds while avoiding their pitfalls. Specifically, in actor-critic approaches, the critic component performs policy evaluation to estimate the value or state-action function for the current policy, while the actor component leverages the policy evaluation information to compute policy gradient to update policy parameters. On one hand, with policy evaluation, actor-critic methods share the similar benefit of effective data usage as in value-based approaches. On the other hand, by exploiting policy gradient information, actor-critic methods are versatile and can easily handle many RL complications in practice (e.g., infinite or continuous state and action spaces). Moreover, thanks to the use of policy gradients, actor-critic methods can incorporate numerous insights and techniques from decentralized gradient-based optimization, making them particularly appealing for decentralized MARL.

To date, while actor-critic decentralized MARL methods have shown great empirical promises in various applications (Foerster et al., 2016; Omidshafiei et al., 2017; Naderializadeh et al., 2021; Feriani and Hossain, 2021; Li et al., 2022), the theoretical foundation of actor-critic decentralized MARL remains in its infancy. Although there have been recent efforts on addressing the theoretical gaps in MARL (Hairi et al., 2022), many fundamental problems remain wide open. There are two major technical limitations in actor-critic decentralized MARL that could significantly diminish the long-term applicability and promises of actor-

---

[*]Equal contribution   [1]The Ohio State University, Columbus, Ohio, USA [2]University of Wisconsin-Whitewater, Whitewater, Wisconsin, USA [3]University of Colorado Boulder, Boulder, Colorado, USA. Correspondence to: Zhiyao Zhang <zhang.15178@osu.edu>, Jia Liu <liu@ece.osu.edu>.

*Proceedings of the $42^{nd}$ International Conference on Machine Learning*, Vancouver, Canada. PMLR 267, 2025. Copyright 2025 by the author(s).

critic MARL if they are not well addressed.

(1) *Critics with Linear Function Approximations:* To date, most existing works on actor-critic MARL methods (e.g., Zhang et al. (2018); Chen et al. (2022); Hairi et al. (2022)) adopt linear functions to approximate value or state-action functions of a given policy. Although the use of linear function approximations is more tractable for theoretical analysis and indeed yield some interesting preliminary understandings of actor-critic MARL methods, it violates the settings of most actor-critic decentralized MARL methods in practice, particularly those with deep-neural-network (DNN)-based critics that are not only nonlinear but also non-convex.

(2) *Stationarity Convergence Guarantees:* So far, most theoretical analyses for actor-critic decentralized MARL methods only ensure the convergence to some stationary solution (Zhang et al., 2018; Hairi et al., 2022; Zhang et al., 2021b), which is a rather weak performance guarantee and merely serves as a necessary condition of local optimality. For actor-critic decentralized MARL, little is known on how to develop algorithms with global optimality convergence guarantee.

All the above technical limitations indicate a significant gap between many empirically well-performing DNN-based actor-critic decentralized MARL methods and the current inadequate theoretical understanding of the design and analysis of the actor-critic decentralized MARL algorithms. Hence, an important question naturally arises:

> **(Q):** Could we develop efficient actor-critic methods for decentralized MARL with DNN-based nonlinear function approximation in the critic component to offer global optimality convergence guarantee?

**2) Technical Challenges:** It turns out that answering the above question is highly nontrivial and involves overcoming at least three major challenges outlined as follows:

(1) Existing actor-critic algorithms for single-agent RL with nonlinear function approximation in critic, which often demand meticulous error control analysis, are inadequate due to the distributed nature of MARL systems. This challenge arises from the computation of global quantities (e.g., advantage functions and global TD-errors) (Zhang et al., 2018; Hairi et al., 2022). While a centralized server can partially mitigate this issue, the lack of direct global information sharing among agents in decentralized MARL makes it exceedingly challenging to adapt single-agent actor-critic RL methods with nonlinear function approximations in MARL.

(2) Even if one employs consensus techniques among decentralized agents to alleviate the above challenge to some degree, the resulting error, compounded by inaccuracies inherent in nonlinear function approximation, could still

lead to difficulty in theoretical analysis. Specifically, compared to decentralized MARL actor-critic methods with linear function approximation in critic (Chen et al., 2022; Hairi et al., 2022), the order between the critic's nonlinear operator and the consensus operator cannot be interchanged, rendering the proof techniques of (Chen et al., 2022; Hairi et al., 2022) ineffective.

(3) Even when only aiming for a stationary point, the aforementioned challenges persist. Furthermore, achieving the global optimum is even more challenging. This is because the gradients derived from the descent lemma, which are sufficient for establishing theory for stationary points, cannot fully capture global information. Toward this end, new techniques are required to handle the updating sequences effectively.

**3) Our Contributions:** The major contribution of this work is that we overcome all the above challenges and develop a new actor-critic algorithm for decentralized MARL with nonlinear function approximation in the critic, which offers global optimality convergence guarantee. To our knowledge, this work takes the first step toward establishing a theoretical foundation for actor-critic decentralized MARL methods by affirmatively answering the above question. Our main technical results are summarized as follows:

- We propose the first DNN-based actor-critic algorithm for fully decentralized MARL problem, which converges to global optimality with a rate of $\mathcal{O}(1/T)$, where $T$ is the number of total iterations. Moreover, we show that, to achieve an $\epsilon$-global-optimal solution, our algorithm enjoys a sample complexity of $\mathcal{O}(1/\epsilon^3)$.

- We note that our theoretical analysis effectively addresses all aforementioned technical challenges. In particular, to tackle the challenges posed by the interplay of decentralization and nonlinearity, we design a *new* technique that maintains pseudo-centralized values to couple decentralized values after nonlinear operation. (cf. Remark 6).

- To verify our theoretical results, we conduct extensive experiments. First, we perform ablation studies in a small-scale MARL environment to demonstrate the impacts of various parameters on the algorithm. Interestingly, our results reveal that the use of TD-error in our algorithm yields significantly better performance than the use of Q-values in existing works. Moreover, we implement our proposed algorithm on a multi-objective alignment problem to demonstrate the effectiveness of our method in RLHF (reinforcement learning from human feedback) for large language models (LLM).

## 2. Related Work

In this section, to put our work into comparative perspectives, we provide overview on two lines of related research:

(1) single-agent actor-critic RL algorithms with DNN-based nonlinear function approximation in critic; and (2) the development of MARL algorithms.

**1) Single-agent Actor-Critic RL Algorithms:** In the vast literature of single-agent RL, the actor-critic framework has received a significant amount of attention. The actor-critic framework combines the strengths of policy gradient methods and sample efficient policy evaluation of value-based methods and has demonstrated impressive capabilities. Recent works (Castro and Meir, 2010; Maei, 2018; Xu et al., 2020a;b) have established and improved the theoretical convergence results of the single-agent actor-critic framework, primarily focusing on linear function approximations. Moreover, recent studies have started to explore the theoretical foundation of actor-critic methods equipped with neural networks (Gaur et al., 2023; 2024; Wang and Hu, 2021). However, the theoretical results of single-agent actor-critic RL cannot be directly extended to decentralized MARL settings due to the inability to compute global quantities in single-agent decentralized settings.

**2) MARL Algorithms:** MARL problem in the tabular setting was first explored by (Littman, 1994; 2001; Lauer and Riedmiller, 2000) for competitive and cooperative settings respectively. In recent years, research has increasingly focused on MARL with function approximation (Arslan and Yüksel, 2016; Zhang et al., 2018; Hairi et al., 2022) to address problems with large state-action space. However, these works have notable limitations as follows: (1) Foerster et al. (2016); Lowe et al. (2017) assumed the presence of a centralized server, thus not suitable for the decentralized MARL; (2) (Zhang et al., 2018; Hairi et al., 2022; Chen et al., 2022; Hairi et al., 2024; Zeng et al., 2022) focused on theoretical studies on decentralized MARL algorithms with linear function approximation assumption, which are often violated in practice since most empirical actor-critic MARL algorithms adopt nonlinear DNNs; (3) While Gupta et al. (2017); Omidshafiei et al. (2017); Foerster et al. (2016) consider deep MARL algorithms, they provide only numerical results without any theoretical finite-time convergence rate analysis. Moreover, most theoretical results in the MARL literature only guarantee convergence to a stationary solution (Zhang et al., 2018; Hairi et al., 2022; Zhang et al., 2021b), while results on achieving global optimality convergence remain very limited. Although (Chen et al., 2022) indeed guarantees global convergence, however, their algorithm is based on linear function approximation. In contrast, we proposes a DNN-based actor-critic algorithm for decentralized MARL with finite-time global optimality convergence guarantee.

## 3. Problem Formulation and Preliminaries

In this section, we first introduce the problem formulation in Section 3.1, which is followed by preliminaries on the deep neural networks we use and the associated critic optimization problem in Section 3.2.

### 3.1. Problem Formulation

**1) System Model:** We model an MARL problem as a graph network $\mathcal{G} = (\mathcal{N}, \mathcal{E})$, where the node set $\mathcal{N} = \{1, 2, \ldots, N\}$ represents the $N$ agents, and the edge set $\mathcal{E}$ specifies the pairs of agents that can directly communicate. The consensus weight matrix associated with graph $\mathcal{G}$ is denoted as $A$. We consider a multi-agent Markov decision process (MAMDP) denoted as $(\mathcal{S}, \{\mathcal{A}^i\}_{i \in \mathcal{N}}, P, \{r^i\}_{i \in \mathcal{N}}, \gamma, \mathcal{G})$, where $\mathcal{S}$ is the state space, $\mathcal{A}^i$ is local action space for agent $i$, $\mathcal{A} = \prod_{i \in \mathcal{N}} \mathcal{A}^i$ is the joint action space, $P : \mathcal{S} \times \mathcal{A} \times \mathcal{S} \to [0, 1]$ is the global transition matrix, $r^i : \mathcal{S} \times \mathcal{A}^i \to \mathbb{R}$ is agent $i$'s local reward, and $\gamma \in (0, 1)$ is the discount factor. Notably, we consider a "restart" kernel defined as $P(s, a, s') = \gamma \mathbb{P}(s'|s, a) + (1 - \gamma)\mathbb{I}\{s' = s_0\}$, where $s_0$ is the initial state, and $\mathbb{I}\{\cdot\}$ denotes the indicator function that outputs 1 if the event holds, and 0 otherwise. As shown in Section 4, this *restart* kernel significantly simplifies the gradient computation. In MAMDP, each agent $i$ follows a local policy $\pi^i$ parameterized by $\theta^i$ to determine its actions. The joint policy is denoted as $\pi = \prod_{i \in \mathcal{N}} \pi^i$, with the corresponding global parameter $\theta = ((\text{vec}(\theta^1))^\top, \ldots, (\text{vec}(\theta^N))^\top)^\top$, where $\text{vec}(\cdot)$ turns the parameter $\theta^i$ into a vector.

**2) Problem Statement:** We consider a non-competing setting, where all $N$ agents cooperatively maximize the total rewards. Specifically, a MAMDP models a sequential decision-making process with the aim to maximize the long-term discounted cumulative system-wide reward as follows:

$$J(\theta) = \mathbb{E}_{\pi_\theta}\left(\sum_{t=0}^{\infty} \frac{\gamma^t}{N} \sum_{i \in \mathcal{N}} r_{t+1}^i\right) = \mathbb{E}_{\pi_\theta}\left(\sum_{t=0}^{\infty} \gamma^t \bar{r}_{t+1}\right), (1)$$

where $\bar{r}_t = \frac{1}{N}\sum_{i \in \mathcal{N}} r_t^i$. The goal of the MAMDP is to find an optimal policy $\theta^*$ that maximizes $J(\theta)$.

**3) The Actor-Critic Approach.** In this work, we consider using a multi-agent actor-critic algorithm with DNN-based critic to solve the decentralized MARL problem. As noted earlier, the actor-critic framework is well-established in the literature (Sutton, 2018; Mnih et al., 2016; Xu et al., 2020a). This framework alternates between two processes: the critic estimates the value functions for a given policy, and the actor improves the policy based on the critic's policy evaluation result and following policy gradient directions. To compute policy gradients, we first define the state-action value function, i.e., Q-function, associated with the MAMDP is defined as follows:

$$Q_\theta(s, a) := \mathbb{E}\left(\sum_{t=0}^{\infty} \gamma^t \bar{r}_{t+1} \Big| s_0 = s, a_0 = a, \pi_\theta\right). \quad (2)$$

Intuitively, $Q_\theta(s, a)$ represents the "goodness" of the state-action pair $(s, a)$ under policy $\pi_\theta$. However, the exact Q-function values are unavailable during learning. Thus, we use a DNN parameterized by $W$ (cf. Section 3.2) as the model of $\hat{Q}(s, a; W)$, which approximates the $Q_\theta(s, a)$ value. To bootstrap the Q-function estimation, we define the Bellman operator $\mathcal{T}^{\pi_\theta}$ for policy $\pi_\theta$ as follows:

$$\mathcal{T}^{\pi_\theta}\hat{Q}(s, a; W) = \mathbb{E}_\theta\Big(\bar{r}(s, a) + \gamma\hat{Q}(s', a'; W)\Big), \quad (3)$$

where expectation $\mathbb{E}_\theta$ is taken over $s' \sim P(\cdot|s, a), a' \sim \pi_\theta(\cdot|s')$. Since $Q_\theta$ is the fixed point of $\mathcal{T}^{\pi_\theta}$, we define an optimization problem to estimate $Q_\theta(s, a)$ in Section 3.2. In addition, we also define the following advantage function:

$$\text{Adv}_\theta(s, a) = Q_\theta(s, a) - \mathbb{E}_{a\sim\pi_\theta}\left(Q_\theta(s, a)\right). \quad (4)$$

### 3.2. Preliminaries

**1) Deep Neural Networks (DNN):** As mentioned earlier, most of the existing works on actor-critic MARL methods rely on linear function approximations for policy evaluation (i.e., critic) (Chen et al., 2022; Zhang et al., 2018; 2021b; Hairi et al., 2022). However, linear function approximations are not rich enough for complex MARL scenarios. To address this limitation, we adopt DNNs in critic and actor to approximate Q-functions and represent policies, respectively.

Specifically, each agent $i$'s critic maintains a DNN-based local approximation $\hat{Q}(\cdot; W^i)$ to approximate $Q_\theta$, where $W^i$ denotes the weights of critic DNN, which is of width $m$ and depth $D$. For any state-action pair $(s, a) \in \mathcal{S} \times \mathcal{A}$, we use a one-to-one mapping $\xi$ such that $x = \xi(s, a) \in \mathbb{R}^d$. Without loss of generality, we assume $\|x\|_2 = 1$. For simplicity, we use $(s, a)$ and $x$ interchangeably throughout the rest of the paper. The structure of the DNN is as follows:

$$\begin{aligned} x^{(0)} &= Hx, y = b^\top x^{(D)}, \\ x^{(h)} &= \frac{1}{\sqrt{m}}\text{ReLU}(W^{(h)}x^{(h-1)}) \text{ for any } h \in [D], \end{aligned} \quad (5)$$

where $[D] = \{1, \ldots, D\}$, $H \in \mathbb{R}^{m \times d}$, $W^{(h)} \in \mathbb{R}^{m \times m}$, and $b \in \mathbb{R}^m$ are the parameters of the DNN, $\text{ReLU}(x) := \max\{0, x\}$ is the ReLU activation function. To initialize parameters, we let all entries of $H$ and $W^{(h)}, \forall h \in [D]$ follow $\mathcal{N}(0, 2)$ independently, and those in $b$ follow $\mathcal{N}(0, 1)$ independently. During training, only $W = (\text{vec}(W^{(1)})^\top, \ldots, \text{vec}(W^{(D)})^\top)^\top$ is updated while $H$ and $b$ remain fixed. Hence, we simplify the notation $\hat{Q}_\theta(x; W, H, b)$ to $\hat{Q}_\theta(x; W)$.

The DNN structure adopted in this paper is the so-called fully connected network (Sainath et al., 2015; Schwing and Urtasun, 2015), which possesses several important theoretical properties. One key property we leverage is the universal

approximation theorem, which says that under sufficiently large depth or width, a fully connected neural network can accurately approximate any functions (Jacot et al., 2018; Gao et al., 2019; Allen-Zhu et al., 2019; Liu et al., 2024). We adopt the same fully connected network structure in the actor of each agent $i \in \mathcal{N}$, which is similar to (Liu et al., 2020a; Gaur et al., 2024). The only difference in our DNN is the addition of a Softmax layer at the output to represent a probability distribution. The input to the DNN is the current state $s \in \mathcal{S}$. The output of the DNN is an action $a \in \mathcal{A}^i$. For simplicity, we assume $|\mathcal{A}^i|$ to be the same for all $i \in \mathcal{N}$. Consequently, the local policy $\pi_\theta^i$ is parameterized by the vector $\theta^i$ of dimension $m(Dm + d + 1)$.

**2) Optimization Problems for Policy Evaluation:** For a given policy $\pi_\theta$, since $Q_\theta$ is the fixed point of Bellman operator defined in Eq. (3), we can find $Q_\theta$ by solving the following minimization problem:

$$\text{MSBE}(W) = \mathbb{E}_{x\sim\nu(\theta)}\left[(\hat{Q}(x; W) - \mathcal{T}^{\pi_\theta}\hat{Q}(x; W))^2\right], (6)$$

where $x$ follows the stationary distribution $\nu(\theta)$, which will be introduced in Lemma 5.4. Eq. (6), referred to as the mean-squared Bellman error (MSBE) (Cai et al., 2019), quantifies the gap between the approximation $\hat{Q}$ and the true fixed point $Q_\theta$ under parameter $W$. A surrogate of MSBE is the projected mean-squared Bellman error (MSPBE) operator, which is defined as:

$$\text{MSPBE}(W) = \mathbb{E}_{x\sim\nu(\theta)}\left[(\hat{Q}(x; W) - \Pi_\mathcal{F}\mathcal{T}^{\pi_\theta}\hat{Q}(x; W))^2\right], (7)$$

where $\Pi_\mathcal{F}$ is the projection map onto the function class $\mathcal{F}$.

## 4. The DNN-Based Actor-Critic Method for Decentralized MARL

**1) Algorithm Overview:** Our proposed DNN-based actor-critic method for decentralized MARL is presented in Algorithm 1 with Algorithm 2 serving as its critic component. The overall algorithm in Algorithm 1 employs a double-loop structure. The inner-loop, corresponding to the critic in Algorithm 2, runs $K$ iterations to approximate Q-function using temporal difference (TD) learning. The outer-loop in Algorithm 1 iterates $T$ rounds, where in each round, the policy $\theta$ is updated using the newly obtained Q-function approximation from the inner-loop. Moreover, a gossiping technique (Nedic and Ozdaglar, 2009) is used in both the actor and critic to efficiently broadcast local information. This enables a consensus process and effectively addresses the decentralization challenges of the problem.

**2) The Critic Component:** Algorithm 2 shows the critic component of our algorithm. Given the current policy $\pi_{\theta_t}$, each agent $i \in \mathcal{N}$ leverages TD-learning to approximate $Q_{\theta_t}$. Specifically, in each iteration $k$ of the total $K$ iterations,

**Algorithm 1** DNN-based Actor-Critic for Dec. MARL.

1: **Input:** step-size $\alpha_t$, initial parameters $\theta_0^i$ for all $i \in \mathcal{N}$.
2: **for** $t = 0, 1, \ldots, T-1$ **do**
3:     Let $W_t, s_{t,0}$ be output of Algorithm 2.
4:     **for** $l = 0, 1, \ldots, M-1$ **do**
5:         **for** $i \in \mathcal{N}$ **do**
6:             Observe: $s_{t,l+1}$ and $r_{t,l+1}^i$.
7:             Sample: $a_{t,l+1}^i \sim \pi_{\theta_t^i}^i(\cdot|s_{t,l+1})$.
8:             Compute: $\psi_{t,l}^i = \nabla_{\theta^i} \log \pi_{\theta_t^i}^i(s_{t,l}, a_{t,l}^i)$.
9:             Compute $\delta_{t,l}^i = \hat{Q}(s_{t,l}, a_{t,l}; W_t^i) - r_{t,l+1}^i - \gamma \hat{Q}(s_{t,l+1}, a_{t,l+1}; W_t^i)$.
10:         **end for**
11:     **end for**
12:     Stack $\tilde{\Delta}_0 = \begin{bmatrix} \delta_{t,0}^1 & \cdots & \delta_{t,0}^N \\ \vdots & \ddots & \vdots \\ \delta_{t,M-1}^1 & \cdots & \delta_{t,M-1}^N \end{bmatrix}$.
13:     Compute $\tilde{\Delta} = A^{t_{\text{gossip}}} \tilde{\Delta}_0^\top$.
14:     **for** $i \in \mathcal{N}$ **do**
15:         Assign $\tilde{\delta}_{t,:}^i := \tilde{\Delta}(i,:)^\top$.
16:         Compute: $d_t^i = \frac{1}{M} \sum_{l=0}^{M-1} \tilde{\delta}_{t,l}^i \psi_{t,l}^i$.
17:         Update: $\theta_{t+1}^i = \theta_t^i + \alpha_t \frac{d_t^i}{\|d_t^i\|}$.
18:     **end for**
19: **end for**
20: **Output:** $\theta_T^i$ for all $i \in \mathcal{N}$ (i.e., $\theta_T$).

---

**Algorithm 2** DNN-Based Critic for Decentralized MARL.

1: **Input:** $s_0$, $\pi_{\theta_t}$, step-size $\beta$, iteration number $K$, consensus weight matrix $A$, gossiping times $t_{\text{gossip}}$.
2: **Initialize:** $\mathcal{B}(B) = \{W : \|W^{(h)} - W^{(h)}(0)\|_{\text{F}} \leq B, \forall h \in [D]\}$. $W^i(0) = W^i = W(0), \forall i \in \mathcal{N}$.
3: **for** $k = 0, 1, \ldots, K-1$ **do**
4:     **for** $i \in \mathcal{N}$ **do**
5:         Sample the tuple: $(s_k, a_k^i, r_{k+1}^i, s_{k+1}, a_{k+1}^i)$, where $a_k^i \sim \pi_{\theta^i}^i(\cdot|s_k)$.
6:         Compute TD-error: $\delta_k^i = \hat{Q}(s_k, a_k; W^i(k)) - r_{k+1}^i - \gamma \hat{Q}(s_{k+1}, a_{k+1}; W^i(k))$.
7:         Update: $\tilde{W}^i(k+1) = W^i(k) - \beta \delta_k^i \cdot \nabla_W \hat{Q}(s_k, a_k; W^i(k))$.
8:         Project: $W^i(k+1) = \arg\min_{W \in \mathcal{B}(B)} \|W - \tilde{W}^i(k+1)\|_2$.
9:         Update $W^i$: $W^i = \frac{k+1}{k+2} W^i + \frac{1}{k+2} W^i(k+1)$.
10:     **end for**
11: **end for**
12: Gossip: Set $\hat{W} = (W^1, \ldots, W^N)^\top$, $W_K = A^{t_{\text{gossip}}} \hat{W}$.
13: **Output:** $s_{K-1}, W_K$.

---

agents first implement Markovian sampling according to policy $\pi_{\theta_t}$. Then, each agent $i$ computes the local TD-error $\delta_k^i$ and updates the parameter $\tilde{W}^i(k+1)$ based on $\delta_k^i$.

Importantly, after each TD learning update, the parameter is projected onto a projection ball with a radius of $B > 0$ and centered on the global initial parameter $W^i(0) = W(0)$. This projection step ensures a non-expansive property that will be useful in our subsequent theoretical analysis.

Lastly, since all $N$ agents update their parameters based on local data, upon the completion of $K$ local update iterations in critic, each agent performs the consensus process using the gossiping technique (communicate only with local neighbors and perform local weighted aggregations) to aggregate information from neighboring agents. The gossiping process will be executed for $t_{\text{gossip}}$ rounds. This iterative weighting process ensures reaching a near consensus on the average information across all agents (Nedic and Ozdaglar, 2009; Zhu et al., 2021; Hairi et al., 2022; Zhang et al., 2021a). To state the gossiping process, we collect all aggregation weights and form a consensus matrix $A$ as follows:

**Definition 4.1** (Consensus Matrix). A consensus matrix $A \in \mathbb{R}^{N \times N}$ associated with the graph $\mathcal{G} = \{\mathcal{N}, \mathcal{E}\}$ is defined as follows: (i) The $(i,j)$-th element $A_{ij} = 0$ if $(i,j) \notin \mathcal{E}$, implying agents $i$ and $j$ cannot share information directly; and (ii) $A_{ij} > 0$ denotes the weight between two agents when they communicate with each other.

Using the consensus matrix $A$, the consensus process across all agents can be compactly written in matrix form as $W_K = A^{t_{\text{gossip}}} \hat{W}$, where $\hat{W} := (W^1, \ldots, W^N)^\top$.

**3) The Actor Component:** As shown in Algorithm 1, the actor component runs $T$ iterations. In each iteration $t$, agents first receive the updated consensual parameters from the critic. Subsequently, a Markovian batch sampling of length $M$ is employed. In this process, agents maintain the local TD-error $\delta_{t,l}^i$ and the local score function $\psi_{t,l}^i$ using the newly gathered Markovian data. This information is used in computing the MARL policy gradient.

However, since the policy gradient computation is based on the visitation measure of the policy as shown in (Sutton, 2018), directly evaluating the policy gradient is challenging. Fortunately, thanks to the *restart* kernel, we obtain the following key result that significantly simplifies the policy gradient computation.

**Lemma 4.2.** *Denote $\eta(s) := \sum_{t=0}^{\infty} \gamma^t \mathbb{P}_{\pi_\theta}(s_0 \to s, t)$ as the visitation measure, where $\mathbb{P}_{\pi_\theta}(s_0 \to s, t)$ is the probability of visiting state $s$ after $t$ steps under policy $\pi_\theta$, with starting state $s_0$. Then, under the restart kernel, the stationary distribution $\nu(\cdot)$ is proportional to $\eta(\cdot)$. More specifically, for any state $s$, we have $\nu(s) = (1-\gamma)\eta(s)$.*

We also note that the *restart* kernel not only provides the elegant "proportional" result above, but is has also widely adopted in the RL literature (Konda and Tsitsiklis, 2003; Chen et al., 2022; Xu et al., 2020a). Based on Lemma 4.2, we can rewrite the policy gradient theorem as follows. Due

to space limitation, we relegate the proofs of Lemma 4.2 and Lemma 4.3 to Appendix B.

**Lemma 4.3** (Policy Gradient Theorem for MARL). *For any parameter $\theta$ of policy $\pi_\theta$: $\mathcal{S} \times \mathcal{A} \to [0,1]$, the gradient of $J(\theta)$ with respect to local parameter $\theta^i$ can be written as:* $\nabla_{\theta^i} J(\theta) \propto \mathbb{E}_{s \sim \nu(\theta), a \sim \pi_\theta}[\nabla_{\theta^i} \log \pi_{\theta^i}^i(a^i|s) \cdot Adv_\theta(s,a)]$.

However, the local TD-error $\delta_{t,l}^i$ cannot be directly used in Lemma 4.3 since advantage functions, which represent the expectation of TD-errors, are defined over the global state-action pairs. To address this problem, we use the gossiping technique again to derive the consensual TD-error $\tilde{\delta}_{t,l}^i$ (cf. Line 12 in Algorithm 1). Lastly, the policy parameter $\theta_{t+1}$ is updated using the gradient descent method.

**Remark 1.** It is worth pointing out that Lemma 4.3 has an alternative form, where the advantage function $Adv_\theta(s,a)$ is replaced by the Q-function $Q_\theta(s,a)$. This approach has been adopted in numerous studies (Sutton et al., 1999; Lowe et al., 2017; Cai et al., 2019; Gaur et al., 2024; Szepesvári, 2022). However, our numerical results in Section 6.1 show that the empirical performance of using $Adv_\theta(s,a)$ significantly better than that of using $Q_\theta(s,a)$.

**Remark 2.** Since our Algorithms 1 and 2 compute Q-functions, global state and global action are needed in the algorithms. We note that this requirement is not always restrictive and has also been adopted in the literature (Zhang et al., 2018; Zeng et al., 2022; Wai et al., 2018). As a concrete example, when using MARL for optimizing the performance of the carrier sensing multiple access (CSMA) protocol of wireless random access networks, participating devices (agents) can listen to packet signals transmitted over the channel and be aware of the actions other devices have taken. In general, applications may include any system where the agents can observe one another for information sharing, including robotics, drones, vehicles, and beyond.

## 5. Theoretical Convergence Analysis

In this section, we start by introducing some assumptions and associated supporting lemmas, followed by the main theoretical results and important remarks.

**Assumption 5.1** (Consensus Matrix). The consensus matrix $A$ is doubly stochastic and there exists a constant $\eta > 0$ such that $A_{ii} \geqslant \eta, \forall i \in \mathcal{N}$, and $A_{ij} \geqslant \eta$ if $(i,j) \in \mathcal{E}$.

Assumption 5.1 is widely adopted in decentralized applications (Hairi et al., 2022; Nedic and Ozdaglar, 2009; Zhu et al., 2021; Oh et al., 2025). This assumption also ensures that $A^\tau$ remains doubly stochastic for any $\tau > 0$.

**Assumption 5.2** (Markov Chain). For any $s \in \mathcal{S}$, $a^i \in \mathcal{A}^i, \forall i \in [N]$, and $\theta^i, \forall i \in [N]$, suppose $\pi_{\theta^i}^i(a^i|s) \geqslant 0$, and $\pi_{\theta^i}^i(a^i|s)$ is differentiable w.r.t. $\theta^i$. In addition, the Markov chain $\{s_t\}, t \geqslant 0$ is irreducible and aperiodic.

**Assumption 5.3** (Bounded Reward). There exists a positive constant $r_{\max}$, such that $0 \leqslant r_t^i \leqslant r_{\max}$ for any $t \geqslant 0, i \in \mathcal{N}$.

Assumption 5.2 and Assumption 5.3 are standard in MARL setting (Zhang et al., 2018; Hairi et al., 2022; Xu et al., 2020a). The former implies the existence of a unique stationary distribution $\nu(\theta)$, while the latter ensures a uniform upper bound for all instantaneous rewards. The long-term mixing time behavior of MAMDP is shown as follows:

**Lemma 5.4** (Mixing Time). *Suppose Assumption 5.2 holds. There exist a stationary distribution $\nu$ for $(s,a)$, and positive constants $\kappa$ and $\rho \in (0,1)$, such that $\sup_{s \in \mathcal{S}} \|P(s_t, a_t|s_0 = s) - \nu(\theta)\|_{TV} \leqslant \kappa \rho^t, \forall t \geqslant 0$.*

**Assumption 5.5.** For any policy $\theta$, and any state-action pair $(s,a)$, there exists a positive constant $\mu_f$ such that the score function satisfies: $\|\nabla_\theta \log \pi_\theta(a|s)\| \leqslant 1$, $\mathbb{E}_{\nu(\theta)}(\nabla_\theta \log \pi_\theta(a|s) \nabla_\theta \log \pi_\theta(a|s)^\top) \succeq \mu_f I$, where $(s,a) \sim \nu(\theta)$, and $\succeq$ denotes semi-positive definite.

**Assumption 5.6.** For any policy $\theta$, there exists a positive constant $\epsilon_{\text{bias}}$ such that: $\mathbb{E}\big(Adv_\theta(s,a) - (1 - \gamma)(F(\theta)^\dagger \nabla_\theta J(\theta))^\top \nabla_\theta \log \pi_\theta(a|s)\big)^2 \leqslant \epsilon_{\text{bias}}$, where $(s,a)$ follows the visitation distribution under optimal policy $\pi^*$, $F(\theta) = \nabla_\theta \log \pi_\theta(a|s) \nabla_\theta \log \pi_\theta(a|s)^\top$, and $\dagger$ denotes the pseudo-inverse of the matrix.

Assumption 5.5 implies the Fisher-information matrix is non-degenerate (Yuan et al., 2022; Fatkhullin et al., 2023), and Assumption 5.6 indicates that the advantage function can be approximated by using the score function (Ding et al., 2022; Agarwal et al., 2021). Note that our parameterization guarantees the use of Gaussian policies, and (Fatkhullin et al., 2023) further suggests that Assumption 5.5 holds in many scenarios. These two assumptions provide the following key lemma (Agarwal et al., 2021), which reveals the relationship between **global optimality** gap and the gradient of $J$, which is stated as follows:

**Lemma 5.7.** *Under Assumption 5.5 and Assumption 5.6, for any policy $\theta$, it holds that $\sqrt{\mu}(J(\theta^*) - J(\theta)) \leqslant \epsilon' + \|\nabla_\theta J(\theta)\|$, where $\mu = \frac{\mu_f^2}{2}$, and $\epsilon' = \frac{\mu \sqrt{\epsilon_{bias}}}{1-\gamma}$.*

**Assumption 5.8** (Lipschitz Continuity). $J(\theta)$ and $\hat{Q}_\theta(x; W, A, b)$ are $L_J$- and $L_W$-Lipschitz continuous w.r.t. $\theta$ and $W$, respectively, i.e., there exist some positive constants $L_J$ and $L_W$ such that, for any $\theta$ and $\theta'$, and for any $W$ and $W'$, we have:

$$|J(\theta) - J(\theta')| \leqslant L_J \|\theta - \theta'\|_2,$$
$$|\hat{Q}_\theta(x; W, H, b) - \hat{Q}_\theta(x; W', H, b)| \leqslant L_W \|W - W'\|_2.$$

Assumption 5.8, known as the smoothness condition, is also standard in the literature (Fatkhullin et al., 2023; Gaur et al., 2024; Xu et al., 2020a; Hairi et al., 2022). This makes it possible to establish the descent lemma in analysis.

**Assumption 5.9** (Universal Approximation). Suppose for any policy $\theta$, the critic estimation $\hat{Q}(\cdot; W^*)$, where $W^*$ is called a stationary point and is defined in Fact C.3, can be close enough to the ground truth $Q_\theta(\cdot)$, i.e., there exists a positive constant $\epsilon_{\text{critic}}$ such that: $\max_{\theta,x} |\hat{Q}(x; W^*) - Q_\theta(x)| \leqslant \epsilon_{\text{critic}}$.

The above assumption describes the approximation quality of critic networks: for any policy $\theta$, the optimal fully connected neural network is able to accurately approximate the Q-function for any state-action pair $(s, a)$.

Now we are ready to present the main result while deferring the detailed proof to Appendix C due to space limitations.

**Theorem 5.10.** *Under all stated assumptions, by selecting $\alpha = \frac{7}{2\sqrt{\mu}}$, $\alpha_t = \frac{\alpha}{t}$, $\beta = 1/\sqrt{K}$, $m = \Omega(d^3 D^{-\frac{11}{2}})$, $B = \Theta(m^{\frac{1}{32}} D^{-6})$, and $K = \Omega(D^4)$, then with probability at least $1 - \exp(-\Omega(\log^2 m))$, Algorithm 1 satisfies:*

$$\mathbb{E}\left(J(\theta^*) - J(\theta_T)\right)$$

$$= \underbrace{\mathcal{O}\left(T^{-1}\right)}_{(1)} + \underbrace{\mathcal{O}\left(\sqrt{\epsilon_{bias}}\right)}_{(2)} + \underbrace{\mathcal{O}\left(\epsilon_{critic}\right)}_{(3)}$$

$$+ \underbrace{\mathcal{O}\left(N^{\frac{1}{2}} M^{-\frac{1}{2}}\right)}_{(4)} + \underbrace{\widetilde{\mathcal{O}}\left(N^{\frac{3}{2}} m^{\frac{1}{32}} D^{-\frac{11}{2}} (1 - \eta^{N-1})^{t_{gossip}}\right)}_{(5)}$$

$$+ \underbrace{\widetilde{\mathcal{O}}\left(N^{\frac{1}{2}} m^{\frac{1}{32}} D^{-\frac{11}{2}}\right)}_{(6)}$$

$$+ \underbrace{\widetilde{\mathcal{O}}\left(N^{\frac{1}{2}} m^{\frac{1}{32}} D^{-\frac{7}{2}} K^{-\frac{1}{4}}\right) + \widetilde{\mathcal{O}}\left(N^{\frac{1}{2}} m^{-\frac{1}{24}} D^{-4}\right)}_{(7)}.$$

The following results immediately follow from Theorem 5.10:

**Corollary 5.11** (Sample and Communication Complexity). *To achieve an $\epsilon$-accurate point, i.e., to ensure $\mathbb{E}\left(J(\theta^*) - J(\theta_T)\right) \leqslant \epsilon + \mathcal{O}(\epsilon_{critic} + \sqrt{\epsilon_{bias}})$, for any constant $\epsilon > 0$, setting $T = \Omega(\epsilon^{-1})$, $M = \Omega(N\epsilon^{-2})$, and $K = \Omega(N^{\frac{1}{2}} \epsilon^{-1})$ yields the following sample complexity $T(NK + NM) = \mathcal{O}(\frac{N^2}{\epsilon^3})$. In addition, selecting $t_{gossip} = \Omega(\log \frac{N^{\frac{3}{2}}}{\epsilon})$ yields the following communication complexity: $2T t_{gossip} = \mathcal{O}(\frac{1}{\epsilon} \log \frac{N^{\frac{3}{2}}}{\epsilon})$.*

Several important remarks on Theorem 5.10 are in order:

**Remark 3.** To elucidate the insights of Theorem 5.10, we derive each term in detail. **a)** Terms (1) and (2) stem from the actor's iterations, and can be derived iteratively by Lemma 5.7. The first term diminishes as the total number of iterations increases, while the second term remains consistently small due to the superior approximation capability of neural networks. **b)** The remaining terms are

associated with controlling the gap between the update direction $d_t$ and true gradient $\nabla J(\theta_t)$ in every iteration $t$. Specifically, to bridge $d_t$ and $\nabla J(\theta_t)$, we introduce an intermediate direction $d_t^*$, derived from $W_{\theta_t}^*$ instead of local $W_t^i$'s to compute TD-errors, and aim to bound $\|d_t - d_t^*\|^2$ and $\|d_t^* - \nabla J(\theta_t)\|^2$. On the one hand, $\|d_t^* - \nabla J(\theta_t)\|^2$ can be decomposed into three components: the limitations of optimal network's representational capacity, the gap between TD-error and advantage function, and the error caused by insufficient gossiping in actor. These components ultimately give rise to terms (3), (4) and (5), respectively. On the other hand, $\|d_t - d_t^*\|^2$ is even more tricky to handle. To this end, averaged $\bar{W}$ and pseudo-centralized $\bar{V}$ are delicately introduced as extra intermediate terms. Ultimately, terms (5), (6), and (7) are used to control this term. Overall, it is worth noting that all these terms can be effectively controlled by increasing the network size, the number of gossip iterations, and the batch size of Markov sampling.

**Remark 4.** Theorem 5.10 demonstrates that Algorithm 1 converges to the neighborhood of the global optima at a rate of $\mathcal{O}(1/T)$. Compared to the state-of-the-art MARL actor-critic algorithm with linear approximation (Hairi et al., 2022), our convergence rate remains the same, highlighting that our carefully crafted design and analysis effectively address the challenges of non-linearity. Furthermore, our approach achieves **global optimality**, as opposed to merely converging to a stationary point measured by $\|\nabla J(\theta)\|^2$. This global convergence guarantee follows from Lemma 5.7 and the subsequent precise control. Specifically, combining with descent lemma, Lemma 5.7 transforms the task of handling iterative relations between consecutive steps into bounding the gap between $J(\theta_t), \forall t$ and the **global optimal policy** $J(\theta^*)$. We note that our analysis techniques in proving global optimality could be of independent interests in the RL literature.

**Remark 5.** Compared to (Gaur et al., 2024), which introduces an actor-critic framework for single-agent RL scenarios with DNNs, we note the following facts: (1) Unlike the i.i.d. sampling assumption in (Gaur et al., 2024), our method operates under the more realistic **Markov sampling** setting and effectively addresses the associated noise. (2) While (Gaur et al., 2024) claims a $\tilde{\mathcal{O}}(D^{\frac{7}{2}} m^{-\frac{1}{12}})$ bound for the global error, this result does not align well with empirical observations: deeper networks often yield better performance. In contrast, Theorem 5.10 reveals that the dominant orders for the width and depth of the neural networks are $D^{-4}$ and $m^{-\frac{1}{24}}$, respectively, which closes this gap. Our *improved* bound reveals an important insight that **increasing depth of DNN will significantly improves the algorithm's performance, while varying width has little impact on the overall performance.**

**Remark 6.** With an increase in the number of iterations $t_{gossip}$ in the gossiping technique in Algorithm 2, (i.e., ap-

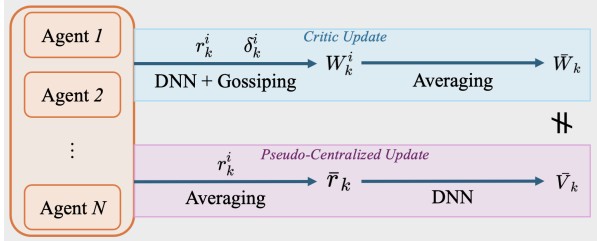

*Figure 1.* The interplay of decentralization and nonlinearity.

plying the consensus matrix $A$ on the parameter matrix $\hat{W}$ sufficiently many times), all agents' parameters will reach a new consensus. However, we note that this consensus process is performed after each agent's nonlinear DNN learning, which yields a *systematic bias* even with an infinite number of gossiping rounds. To address this challenge, we develop a new technique to carefully analyze this bias.

Specifically, for any given policy $\theta$, Assumption 5.9 gives optimal approximation $W_\theta^*$, and we aim to bound the gap between each parameter $W_K^i = W_K(i,:)^\top, \forall i \in \mathcal{N}$ and $W_\theta^*$. After $t_{\text{gossip}}$ iterations of consensus, $W_K^i$ is already close to $\bar{W} = \sum_{i \in \mathcal{N}} W_K^i / N$, and our next goal is to characterize the difference between $\bar{W}$ and $W_\theta^*$. Unfortunately, since the order between the $K$ iterations of **nonlinear operations** and the gossiping process **cannot be directly interchanged** in our proof, we cannot view the results obtained from gossiping as simple averages of the raw local. This fact highlights the **hardness** in establishing the performance of nonlinear actor-critic and its **fundamental difference** compared to linear actor-critic. To see the difference more clearly, as shown in (Hairi et al., 2022), when using linear approximation, the linearity enables a direct handling of the relationships between $W_K^i$ and $W_\theta^*$.

To overcome the above nonlinear challenge, our **key idea** is to maintain a *pseudo-centralized* updating parameter $\bar{V}$, which collects local $r_i$ to obtain $\bar{r}$, and applies it to directly derive centralized TD-error, to bridge the gap between $\bar{W}$ and $W_\theta^*$ (See Fig. 1 and Appendix C for more details).

# 6. Numerical Experiments

In Section 6.1, we first use a simple environment to conduct ablation study to explore the impacts of each algorithmic element. Then, in Section 6.2, we conduct a large-scale experiment with LLM to illustrate our method's efficacy when applied in RLHF.

## 6.1. Ablation Study with the Simple Spread

**1) Experiment Settings:** We conduct ablation studies to verify Theorem 5.10. For this experiment, we consider a modified version of `Simple Spread` (Lowe et al., 2017), a toy environment widely used in MARL. The details of setting and our modifications are provided in Appendix D.1.

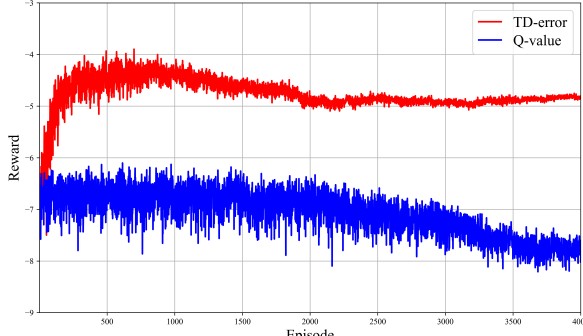

*Figure 2.* Algorithm 1 performed with TD-error and Q-value.

**2) Experiment Results:** Fig. 2 presents our findings on the difference between (1) using the consented TD-error, as shown in Line 16 of Algorithm 1, and (2) replacing this TD-error with the Q-value, as shown in (Lowe et al., 2017; Cai et al., 2019; Gaur et al., 2024). Our algorithm, which incorporates the consensual TD-error, is highly efficient, as agents achieve increasing rewards over time. In contrast, the Q-value-based method fails to exhibit meaningful learning behavior, with rewards even decreasing over episodes. Notably, when using the Q-value approach, the gossiping technique in the actor step is theoretically unnecessary, potentially reducing computational costs. However, our numerical results suggest that this theoretical advantage is offset by the Q-value-based approach's poorer overall performance. More numerical results of ablation studies, which verify Theorem 5.10, can be found in Appendix D.1.

## 6.2. LLM Alignment via Multi-Agent RLHF

**1) Experiment Settings:** To further validate our proposed algorithm, we conduct a large-scale experiment with LLM to test our algorithm's performance in multi-agent RLHF (Ouyang et al., 2022). The system architecture of our multi-agent RLHF framework is shown in Figure 3, where there are multiple local LLMs, each paired with its own reward model (RM) that has been pre-trained from a local dataset. A controller is responsible for communicating with the LLMs and initiating a dialogue. Once the controller initiates a conversation by generating a prompt (global state), the actor part of each LLM processes it and outputs a response (local action). The response is then evaluated by the RM, which returns a score (local reward) that numerically reflects the quality of response. Based on the obtained score, each LLM executes a learning step and updates its weights. All local responses are aggregated into a final response and then shown to the prompting module, which generates the next prompt (state transition). Detailed experiment settings along with additional notes regarding our multi-agent RLHF implementation are provided in Appendix D.2.

**2) Experiment Results:** Fig. 4 shows reward-versus-episode to illustrate the learning performance of multi-agent

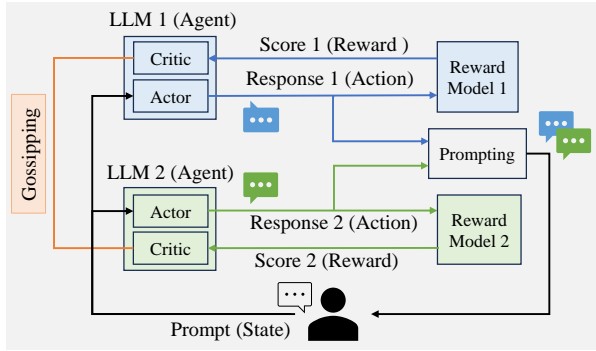

*Figure 3.* A block diagram of multi-agent RLHF with two LLMs.

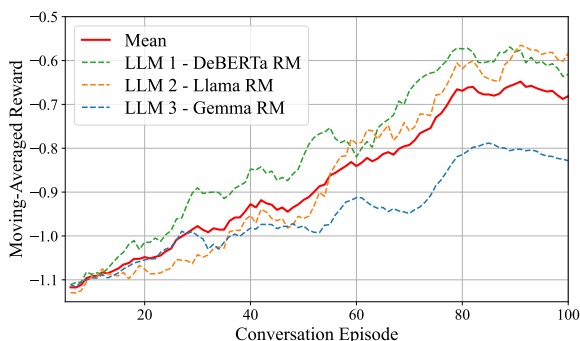

*Figure 4.* A learning performance plot of LLM agents during multi-agent RLHF. Moving average was applied over every 5 episodes.

RLHF. We observe a steady increase in the average reward as the episodes progress, which validates our multi-agent RLHF framework. It is important to note that, despite varying rates of improvement, the reward consistently increases for all LLMs, highlighting the efficacy of our decentralized actor-critic algorithm, effectively leading the agents to a policy that benefits all through the gossiping technique.

## 7. Conclusion

In this paper, we investigated decentralized MARL problems and proposed a DNN-based actor-critic method, which offers the first finite-time convergence guarantees to global optimality, and also enjoys low sample complexity. Our experiments on ablation studies and LLM alignment further verified the effectiveness of our proposed algorithms. Our work contributes to advancing the understanding of MARL actor-critic methods with nonlinear function approximation.

## Acknowledgement

This work is supported in part by NSF grants CAREER 2110259, 2112471, and 2324052; DARPA grants YFA D24AP00265 and HR0011-25-2-0019; ONR grant N00014-24-1-2729; and AFRL grant PGSC-SC-111374-19s.

## Impact Statement

This paper presents work whose goal is to advance the field of Machine Learning. There are many potential societal consequences of our work, none which we feel must be specifically highlighted here.

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

# Appendix

## A. Comparison of Different Algorithms

We summarize the settings and performance comparisons of related algorithms in Table 1.

| Algorithm | Scenario | Approximation | Convergence | Guarantee | Rate |
|---|---|---|---|---|---|
| (Xu et al., 2020a) | SARL | Linear | Global | ✓ | $\mathcal{O}(T^{-1})$ |
| (Gaur et al., 2024) | SARL | Linear | Global | ✓ | $\mathcal{O}(T^{-1})$ |
| (Foerster et al., 2016) | MARL | Nonlinear | Stationary | ✗ | NA |
| (Omidshafiei et al., 2017) | MARL | Nonlinear | Stationary | ✗ | NA |
| (Zhang et al., 2018) | MARL | Linear | Stationary | ✓ | Asymptotic |
| (Hairi et al., 2022) | MARL | Linear | Stationary | ✓ | $\mathcal{O}(T^{-1})$ |
| (Zeng et al., 2022) | MARL | Linear | Stationary | ✓ | $\mathcal{O}(T^{-\frac{2}{5}})$ |
| (Chen et al., 2022) | MARL | Linear | Global | ✓ | $\mathcal{O}(T^{-1})$ |
| **This Work** | **MARL** | **Nonlinear** | **Global** | ✓ | $\mathcal{O}(T^{-1})$ |

*Table 1.* Comparison of Different Algorithms.

## B. Proof of Section 4

### B.1. Proof of Lemma 4.2

*Proof.* We denote the 1-step state transition kernel $\widetilde{P}_{\pi_\theta}(s \to s', 1) = \sum_a P(s, \pi_\theta(a|s), s')$ is written as follows:

$$\widetilde{P}_{\pi_\theta}(s \to s', 1) := \gamma \mathbb{P}_{\pi_\theta}(s \to s', 1) + (1 - \gamma)\mathbb{I}\{s' = s_0\},$$

where $\gamma \in (0,1)$ and $\mathbb{I}\{s' = s_0\}$ represents the indicator function of the event "*the next state is the initial state $s_0$*". In the literature, this kernel is sometimes referred to as the *restart* kernel.

Next, and we will show that, under this restart kernel, the stationary distribution $\nu(s)$ is **proportional** to the visitation measure $\eta(s)$ (Note: to be more rigorous, we call $\eta(s)$ as a "visitation measure" rather than "visitation distribution" here, because it's possible that $\eta(s) > 1$ for some $s$ and hence not being a proper distribution). The proof is as follows:

Note that, for $\gamma < 1$, the visitation measure at state $s'$ is defined as follows and can be written in a recursive form:

$$\eta(s') := \sum_{t=0}^{\infty} \gamma^t \mathbb{P}_{\pi_\theta}(s_0 \to s', t) = \mathbb{I}\{s' = s_0\} + \gamma \sum_s \eta(s)\mathbb{P}_{\pi_\theta}(s \to s', 1).$$

It then follows from the definition of $\eta(s)$ that:

$$\sum_{s'} \eta(s') = \sum_{s'} \sum_{t=0}^{\infty} \gamma^t \mathbb{P}_{\pi_\theta}(s_0 \to s', t) = \sum_{t=0}^{\infty} \sum_{s'} \gamma^t \mathbb{P}_{\pi_\theta}(s_0 \to s', t) = \sum_{t=0}^{\infty} \gamma^t = \frac{1}{1 - \gamma}.$$

Now, we define the following "proper" distribution $\mu(s)$ by normalizing $\eta(s)$:

$$\mu(s) := \frac{\eta(s)}{\sum_{s'} \eta(s')} = (1 - \gamma)\eta(s).$$

In what follows, we will prove that $\mu(\cdot)$ **is indeed the stationary distribution** $\nu(\cdot)$ **under kernel** $\widetilde{P}_{\pi_\theta}$. Hence, the visitation measure is *proportional* to the stationary distribution. As a result, the use of stationary distribution in the policy gradient calculation remains valid. To this end, we first note that:

$$\sum_s \mu(s)\widetilde{P}_{\pi_\theta}(s \to s', 1) = \gamma \sum_s \mu(s)\mathbb{P}_{\pi_\theta}(s \to s', 1) + (1 - \gamma)\sum_s \mu(s)\mathbb{I}\{s' = s_0\},$$

which follows from the definition of $\widetilde{P}_{\pi_\theta}(s \to s', 1)$. Then, by using the definition $\mu(s) := (1-\gamma)\eta(s)$, we can further rewrite the above equation as:

$$\sum_s \mu(s)\widetilde{P}_{\pi_\theta}(s \to s', 1) = (1-\gamma)\left(\gamma\sum_s \eta(s)\mathbb{P}_{\pi_\theta}(s \to s', 1) + \mathbb{I}\{s' = s_0\}\right) = (1-\gamma)\eta(s') = \mu(s').$$

This shows that $\mu(\cdot)$ is the **stationary distribution** $\nu(\cdot)$ **under kernel** $\widetilde{P}_{\pi_\theta}$ and the proof is complete. $\square$

### B.2. Proof of Lemma 4.3

*Proof.* According to Sec. 13.2 in (Sutton, 2018), we know that

$$\nabla_\theta J(\theta) = \sum_s \eta(s)\sum_a \nabla\pi_\theta(a|s)\text{Adv}_\theta(s, a).$$

Multiplying and dividing the right-hand-side by $1-\gamma$ and using $\nu(s) = (1-\gamma)\eta(s)$, we have that

$$\nabla_\theta J(\theta) = \frac{1}{(1-\gamma)}\sum_s (1-\gamma)\eta(s)\sum_a \nabla\pi_\theta(a|s)\text{Adv}_\theta(s, a),$$

which implies that

$$\nabla_\theta J(\theta) \propto \sum_s \nu(s)\sum_a \nabla\pi_\theta(a|s)\text{Adv}_\theta(s, a).$$

$\square$

## C. Proof of Theorem 5.10

*Proof.* Our proof can be separated into three steps: (1) Dealing with actor iterations by induction; (2) Handling the main term of induction; (3) Combining everything together to get the results.

**Step 1: Induct on $J(\theta_t)$.**

We first denote $d_t = [(d_t^1)^\top, \ldots, (d_t^N)^\top]^\top$ as the global update direction in the $t$-th iteration of actor, and $e_t = d_t - \nabla J(\theta_t)$ as the gap between $d_t$ and ground truth of the gradient. According to the first inequality of Assumption 5.8 and Algorithm 1, we have the following descent lemma:

$$\begin{aligned}
-J(\theta_{t+1}) &\leqslant -J(\theta_t) - \langle\nabla J(\theta_t), \theta_{t+1} - \theta_t\rangle + L_J\|\theta_{t+1} - \theta_t\|^2 \\
&\leqslant -J(\theta_t) - \alpha_t\frac{\langle\nabla J(\theta_t), d_t\rangle}{\|d_t\|} + L_J\|\theta_{t+1} - \theta_t\|^2 \\
&\leqslant -J(\theta_t) - \frac{\alpha_t}{3}\|\nabla J(\theta_t)\| + \frac{8\alpha_t}{3}\|e_t\| + L_J\|\theta_{t+1} - \theta_t\|^2,
\end{aligned} \tag{8}$$

where the last inequality can be followed by considering two cases: $\|e_t\| \leqslant \frac{1}{2}\|\nabla J(\theta_t)\|$ and $\|e_t\| > \frac{1}{2}\|\nabla J(\theta_t)\|$. In both cases, we can easily get $\frac{\langle\nabla J(\theta_t), d_t\rangle}{\|d_t\|} \leqslant -\frac{1}{3}\|\nabla J(\theta_t)\| + \frac{8}{3}\|e_t\|$. Then, let $\Delta_t = J(\theta^*) - J(\theta_t)$ be the gap from global optima, and plug Lemma 5.7 into last inequality:

$$\Delta_{t+1} \leqslant \left(1 - \frac{\alpha_t\sqrt{\mu}}{3}\right)\Delta_t + \frac{8\alpha_t}{3}\|d_t - \nabla J(\theta_t)\| + L_J\alpha_t^2 + \frac{\alpha_t}{3}\epsilon'. \tag{9}$$

Now, as the coefficient of $\Delta_t$ is less than 1, we are ready to induct $\Delta_t$ over $t$:

$$\begin{aligned}
\Delta_t &\leqslant \prod_{k=2}^t\left(1 - \frac{\alpha_k\sqrt{\mu}}{3}\right)\Delta_2 \\
&+ \sum_{k=0}^{t-2}\left(\mathbb{I}\{k \geqslant 1\}\prod_{i=0}^{k-1}(1 - \frac{\alpha_{k-i}\sqrt{\mu}}{3})\right)\alpha_{t-k}\left(\|d_{t-k} - \nabla J(\theta_{t-k})\| + \epsilon'\right) \\
&+ L_J\sum_{k=0}^{t-2}\left(\mathbb{I}\{k \geqslant 1\}\prod_{i=0}^{k-1}(1 - \frac{\alpha_{k-i}\sqrt{\mu}}{3})\right)\alpha_{t-k}^2.
\end{aligned} \tag{10}$$

Now, we appropriately select the step-size by letting constant $\alpha = \frac{7}{2\sqrt{\mu}}$, and $\alpha_t = \frac{\alpha}{t}$, inspired by (Gaur et al., 2024). Then, Equation (10) can be further bounded as follows:

$$\Delta_{t+1} \leqslant \frac{1}{t}\Delta_2 + \frac{\alpha}{t}\sum_{\tau=0}^{t-2}\|d_\tau - \nabla J(\theta_\tau)\| + \frac{L_J\alpha^2}{t} + \epsilon'. \tag{11}$$

Since $\epsilon'$ is a constant, and the first term and the second last term converge to 0 in a rate of $1/T$, to bound the RHS, the key step is to control $\|d_t - \nabla J(\theta_t)\|$.

**Step 2: Upper bound $\|d_t - \nabla J(\theta_t)\|$.**

In order to further bound $\Delta_t$ to get the global optimality, we need to upper bound the second term in Equation (11). In this step, we use triangle inequality to decompose $\|d_t - \nabla J(\theta_t)\|$, then we bound each term in decomposition. To this end, we first introduce some facts and notations for the proof later.

**Fact C.1.** *For any vector $v \in \mathbb{R}^{Nm(m+d+1)}$, the squared Euclidean norm can be written as:*

$$\|v\|^2 = \sum_{i \in \mathcal{N}}\|v^i\|^2,$$

*where $v^i \in \mathbb{R}^{m(m+d+1)}$ for any $i \in \mathcal{N}$.*

**Fact C.2.** *Let $\bar{W}_t = \frac{1}{N}\sum_{i \in \mathcal{N}} W_t^i$. Then, according to doubly stochastic $A$ by Assumption 5.1, we have:*

$$\bar{W}_t = \frac{1}{N}\sum_{i \in \mathcal{N}} W_{t,K}^i,$$

*where $W_{t,K}$ is the output of Algorithm 2 for the outer loop indexed by $t$, and $W_{t,K}^i$ is the $i$-th row of $W_{t,K}$, i.e., the agent $i$'s estimated parameter after consensus process (In other words, $W_t^i$ is $W^i$ after $K$ iterations in Algorithm 2 during its $t$-th leveraging in Algorithm 1). Here, $\bar{W}_t$ can be regarded as the maintained pseudo-centralized parameter which is obtained after infinite numbers of consensus iterations.*

**Fact C.3.** *Let locally linear approximated Q-function be:*

$$\hat{Q}_0(x; W) = \hat{Q}(x; W(0)) + \nabla_W \hat{Q}(x; W(0))^\top (W - W(0)), \tag{12}$$

*and corresponding TD-error be:*

$$\delta_0(x, r, x'; W) = \hat{Q}_0(x; W) - r - \gamma\hat{Q}_0(x'; W). \tag{13}$$

*Then, if $W^*$ satisfies:*

$$\mathbb{E}_{(s,a)\sim\nu}\left((\delta_0(x, r, x'; W^*) \cdot \nabla_W \hat{Q}_0(x; W^*))^\top (W - W^*)\right) \geqslant 0, \forall W \in \mathcal{B}(B), \tag{14}$$

*then we say $W^*$ is a stationary point (since there is no descent direction at $W^*$). For any $H$, $W(0)$ and $b$, there exists stationary point $W^*$, and $\hat{Q}_0(\cdot; W^*)$ is the unique, global optimum of the MSPBE corresponding to the projection set $\mathcal{B}(B)$.*

Then, we introduce some notations. For policy $\pi_{\theta_t}$, the corresponding stationary point is denoted as $W_t^*$. To introduce $\tilde{\delta}_{t,l}^i(W_t^*)$ for any $t, l$ and $i$, we first defint the following value:

$$\delta_{t,l}^i(W_t^*) = \hat{Q}(s_{t,l}, a_{t,l}; W_t^*) - r_{t,l+1}^i - \gamma\hat{Q}(s_{t,l+1}, a_{t,l+1}; W_t^*),$$

which replaces $\delta_{t,l}^i$ in Algorithm 1 by using the stationary parameter $W_t^*$ instead of $W_t^i$. Following that, we apply the same consensus process to attain $\tilde{\delta}_{t,l}^i(W_t^*)$. We denote $d_t^{i,*}$ as the direction derived by $\tilde{\delta}_{t,l}^i(W_t^*)$ with the implementation of Algorithm 1, and denote $d_t^* = [(d_t^{1,*})^\top, \ldots, (d_t^{N,*})^\top]^\top$. Then, let $h_t^i(W_t^*) = \frac{1}{M}\sum_{l=0}^{M-1}\delta_{t,l}(W_t^*)\psi_{t,l}^i$, and $h_t(W_t^*) = [h_t^1(W_t^*)^\top, \ldots, h_t^N(W_t^*)^\top]^\top$, where $\delta_{t,l}(W_t^*) = \frac{1}{N}\sum_{i \in \mathcal{N}}\delta_{t,l}^i(W_t^*)$. Similarly, let $g^i(W, \theta) = \mathbb{E}(\hat{\text{Adv}}(s, a; W)\psi_{\theta^i}(s, a^i))$, and $g(W, \theta) = [g^1(W, \theta)^\top, \ldots, g^N(W, \theta)^\top]^\top$, where $\psi_{\theta^i}(s, a) = \nabla_{\theta^i}\log\pi_{\theta^i}^i(s, a^i)$.

Now, we can decompose $\|d_t - \nabla J(\theta_t)\|^2$ now using triangle inequality and Fact C.1:

$$\|d_t - \nabla J(\theta_t)\|^2$$
$$=\|d_t - d_t^* + d_t^* - h_t(W_t^*) + h_t(W_t^*) - g(W_t^*, \theta_t) + g(W_t^*, \theta_t) - \nabla J(\theta_t)\|^2$$
$$\leqslant 6\|d_t - d_t^*\|^2 + 6\|d_t^* - h_t(W_t^*)\|^2 + 6\|h_t(W_t^*) - g(W_t^*, \theta_t)\|^2 + 6\|g(W_t^*, \theta_t) - \nabla J(\theta_t)\|^2 \tag{15}$$
$$\leqslant 6\sum_{i\in\mathcal{N}}\|d_t^i - d_t^{i,*}\|^2 + 6\sum_{i\in\mathcal{N}}\|d_t^{i,*} - h_t^i(W_t^*)\|^2 + 6\sum_{i\in\mathcal{N}}\|h_t^i(W_t^*) - g^i(W_t^*, \theta_t)\|^2 + 6\|g(W_t^*, \theta_t) - \nabla J(\theta_t)\|^2.$$

Therefore, our objective turns to upper bound each term of the RHS of Equation (15).

**Step 2(a): Deal with $\sum_{i\in\mathcal{N}}\|d_t^i - d_t^{i,*}\|^2$.**

To upper bound the sum for all $i \in \mathcal{N}$, we only need to bound each of them. Thus, for any $i \in \mathcal{N}$, we can first rewrite this term as follows:

$$\|d_t^i - d_t^{i,*}\|^2$$
$$=\|\frac{1}{M}\sum_{l=0}^{M-1}\tilde{\delta}_{t,l}^i\psi_{t,l}^i - \frac{1}{M}\sum_{l=0}^{M-1}\tilde{\delta}_{t,l}^i(W_t^*)\psi_{t,l}^i\|^2$$
$$=\|\frac{1}{M}\sum_{l=0}^{M-1}(\tilde{\delta}_{t,l}^i - \tilde{\delta}_{t,l}^i(W_t^*))\psi_{t,l}^i\|^2 \tag{16}$$
$$=\|\frac{1}{M}\sum_{l=0}^{M-1}A^{t_{\text{gossip}}}(i,:)\Big(\tilde{\Delta}_0(l,:) - \tilde{\Delta}_0(W_t^*)(l,:)\Big)\psi_{t,l}^i\|^2,$$

where $\tilde{\Delta}_0(W_t^*)$ is the matrix in where every component $\delta_{t,l}^i(W_t^*)$ replaces $\delta_{t,l}^i$ in $\tilde{\Delta}_0$, and $A^{t_{\text{gossip}}}(i,:)$ denotes the $i$-th line of the matrix $A^{t_{\text{gossip}}}$. These equalities can be easily derived from definitions. We now bound this new form as follows:

$$\|\frac{1}{M}\sum_{l=0}^{M-1}A^{t_{\text{gossip}}}(i,:)\Big(\tilde{\Delta}_0(l,:) - \tilde{\Delta}_0(W_t^*)(l,:)\Big)\psi_{t,l}^i\|^2$$
$$\leqslant \max_l\|A^{t_{\text{gossip}}}(i,:)\Big(\tilde{\Delta}_0(l,:) - \tilde{\Delta}_0(W_t^*)(l,:)\Big)\psi_{t,l}^i\|^2$$
$$=\max_l\left|A^{t_{\text{gossip}}}(i,:)\Big(\tilde{\Delta}_0(l,:) - \tilde{\Delta}_0(W_t^*)(l,:)\Big)\right|^2\|\psi_{t,l}^i\|^2 \tag{17}$$
$$\leqslant \max_l\left|A^{t_{\text{gossip}}}(i,:)\Big(\tilde{\Delta}_0(l,:) - \tilde{\Delta}_0(W_t^*)(l,:)\Big)\right|^2$$
$$\leqslant \max_{i,l}\left|\tilde{\Delta}_0(l,i) - \tilde{\Delta}_0(W_t^*)(l,i)\right|^2,$$

where the second last inequality is due to the first inequality of Assumption 5.5, and the last one is due to the doubly stochastic property of $A$. Since every component in either $\tilde{\Delta}_0$ or $\tilde{\Delta}_0(W_t^*)$ is TD-error, we can further bound the last equation as follows:

$$\left|\tilde{\Delta}_0(l,i) - \tilde{\Delta}_0(W_t^*)(l,i)\right|$$
$$=\left|\hat{Q}(s_{t,l}, a_{t,l}; W_t^i) - r_{t,l+1}^i - \gamma\hat{Q}(s_{t,l+1}, a_{t,l+1}; W_t^i) - \hat{Q}(s_{t,l}, a_{t,l}; W_t^*) + r_{t,l+1}^i + \gamma\hat{Q}(s_{t,l+1}, a_{t,l+1}; W_t^*)\right|$$
$$=\left|\Big(\hat{Q}(s_{t,l}, a_{t,l}; W_t^i) - \hat{Q}(s_{t,l}, a_{t,l}; W_t^*)\Big) + \gamma\Big(\hat{Q}(s_{t,l+1}, a_{t,l+1}; W_t^*) - \hat{Q}(s_{t,l+1}, a_{t,l+1}; W_t^i)\Big)\right| \tag{18}$$
$$\leqslant \left|\hat{Q}(s_{t,l}, a_{t,l}; W_t^i) - \hat{Q}(s_{t,l}, a_{t,l}; W_t^*)\right| + \gamma\left|\hat{Q}(s_{t,l+1}, a_{t,l+1}; W_t^*) - \hat{Q}(s_{t,l+1}, a_{t,l+1}; W_t^i)\right|.$$

On the one hand, note that the two terms share the same form, thus, we only need to find a uniformly upper bound for every state-action pair. On the other hand, to connect $\hat{Q}(s_{t,l}, a_{t,l}; W_t^i)$ and $\hat{Q}(s_{t,l}, a_{t,l}; W_t^*)$, we also need to add and subtract some terms, and leverage triangle inequality. The first one to bridge $W_t^i$ and $W_t^*$ is $\bar{W}_t$, which treats the inaccuracy caused by gossiping and evaluating separately:

$$|\hat{Q}(s_{t,l}, a_{t,l}; W_t^i) - \hat{Q}(s_{t,l}, a_{t,l}; W_t^*)|$$
$$\leqslant |\hat{Q}(s_{t,l}, a_{t,l}; W_t^i) - \hat{Q}(s_{t,l}, a_{t,l}; \bar{W}_t)| + |\hat{Q}(s_{t,l}, a_{t,l}; \bar{W}_t) - \hat{Q}(s_{t,l}, a_{t,l}; W_t^*)|. \tag{19}$$

---

**Algorithm 3** Centralized Critic with Neural Network

---

1: **Input:** $s_0$, $\pi_{\theta_t}$, step-size $\beta$, iteration number $K$, consensus weight matrix $A$, gossiping times $t_{\text{gossip}}$, W(0) from Algorithm 2.
2: **Initialize:** $\mathcal{B}(B) = \{W : \|W^{(h)} - W^{(h)}(0)\|_F \leqslant B, \forall h \in [D]\}$. $\bar{V} = V(0) = W(0)$.
3: **for** $k = 0, 1, \ldots, K - 2$ **do**
4:    Observe the tuple of all $i \in \mathcal{N}$ agents: $(s_k, a_k^i, r_{k+1}^i, s_{k+1}, a_{k+1}^i)$.
5:    Compute TD-error: $\delta_k = \hat{Q}(s_{k,}, a_k; V(k)) - \bar{r}_{k+1} - \gamma \hat{Q}(s_{k+1}, a_{k+1}; V(k))$.
6:    Update: $\tilde{V}(k+1) = V(k) - \beta \delta_k \cdot \nabla_W \hat{Q}(s_k, a_k; V(k))$.
7:    Project: $V(k+1) = \arg\min_{V \in \mathcal{B}(B)} \|V - \tilde{V}(k+1)\|_2$.
8:    Update average $\bar{V}$: $\bar{V} = \frac{k+1}{k+2} \bar{V} + \frac{1}{k+2} V(k+1)$.
9: **end for**
10: **Output:** $\bar{V}$.

---

The first term in Equation (19) can be upper bounded by gossiping techniques. By Assumption 5.8, the definition of $\mathcal{B}(B)$ and gossiping technique, the following inequality:

$$
\begin{aligned}
&|\hat{Q}(s, a; W_t^i) - \hat{Q}(s, a; \bar{W}_t)| \\
\leqslant &L_W \|W_t^i - \bar{W}_t\| \\
= &L_W \|[(A^{t_{\text{gossip}}})_{i,:} W_{t,K}]^\top - \frac{1}{N} W_{t,K}^i\| \\
= &L_W \|\left((A^{t_{\text{gossip}}})_{i,:} - \frac{1}{N} \mathbb{1}^\top\right) W_{t,K}\| \\
\leqslant &L_W \|(A^{t_{\text{gossip}}})_{i,:} - \frac{1}{N} \mathbb{1}^\top\| \cdot \|W_{t,K}\| \\
\leqslant &L_W \mathcal{O}\left(N(1 - \eta^{N-1})^{t_{\text{gossip}}}\right) \cdot \mathcal{O}(BD^{\frac{1}{2}}),
\end{aligned}
\tag{20}
$$

holds for any state-action pair $(s, a)$. In addition, we also need to upper bound $|\hat{Q}(s_{t,l}, a_{t,l}; \bar{W}_t) - \hat{Q}(s_{t,l}, a_{t,l}; W_t^*)|$ in Equation (19). However, $\bar{W}_t$ is actually not the centralized parameter in critic, but the pseudo-centralized parameter. To make it more clear, we introduce the real centralized parameter $\bar{V}_t$, which follows the updating process in Algorithm 3. Therefore, we can apply $\bar{V}_t$ as a bridge to bound the second term in Equation (19). Similarly, we denote $\bar{V}_t$ as the output of Algorithm 3 corresponding to the outer loop indexed by $t$.

Therefore, by triangle inequality, we have:

$$
\begin{aligned}
&|\hat{Q}(s_{t,l}, a_{t,l}; \bar{W}_t) - \hat{Q}(s_{t,l}, a_{t,l}; W_t^*)| \\
\leqslant &|\hat{Q}(s_{t,l}, a_{t,l}; \bar{W}_t) - \hat{Q}(s_{t,l}, a_{t,l}; \bar{V}_t)| + |\hat{Q}(s_{t,l}, a_{t,l}; \bar{V}_t) - \hat{Q}(s_{t,l}, a_{t,l}; W_t^*)|.
\end{aligned}
\tag{21}
$$

For the first term of Equation (21), since both $\bar{W}_t$ and $\bar{V}_t$ belong to $\mathcal{B}(B)$, we can bound it according to Assumption 5.8 and the convexity of $\mathcal{B}(B)$:

$$
\begin{aligned}
&|\hat{Q}(s_{t,l}, a_{t,l}; \bar{W}_t) - \hat{Q}(s_{t,l}, a_{t,l}; \bar{V}_t)| \\
\leqslant &L_W \|\bar{W}_t - \bar{V}_t\| \\
\leqslant &L_W \mathcal{O}(BD^{\frac{1}{2}})
\end{aligned}
\tag{22}
$$

For the second term of Equation (21), we have the following result: When setting $m = \Omega(d^{3/2} \log^{3/2}(m^{3/2}/B) BD^{\frac{1}{2}})$, $B = \mathcal{O}(m^{1/2} D^{-6} \log^{-3} m)$, $\beta = 1/\sqrt{K}$, and $D = \mathcal{O}(K^{1/4})$, with probability at least $1 - \exp^{-\Omega(\log^2 m)}$, we can bound this term as follows (Cai et al., 2019):

$$
\mathbb{E}\left(|\hat{Q}(s_{t,l}, a_{t,l}; \bar{V}_t) - \hat{Q}(s_{t,l}, a_{t,l}; W_t^*)|\right) \leqslant \mathcal{O}\left((BD^{\frac{5}{2}} K^{-\frac{1}{4}} + B^{\frac{4}{3}} m^{-\frac{1}{12}} D^4) \cdot \log^{\frac{3}{2}} m \log^{\frac{1}{2}} K\right).
\tag{23}
$$

Plugging Equations (20), (22) and (23) into Equation (19), we have:

$$
\begin{aligned}
&\mathbb{E}\left(|\hat{Q}(s_{t,l}, a_{t,l}; W_t^i) - \hat{Q}(s_{t,l}, a_{t,l}; W_t^*)|\right) \\
&\leqslant L_W \mathcal{O}\left(N(1 - \eta^{N-1})^{t_{\text{gossip}}}\right) \cdot \mathcal{O}(BD^{\frac{1}{2}}) + L_W \mathcal{O}(BD^{\frac{1}{2}}) \\
&\quad + \mathcal{O}\left((BD^{\frac{5}{2}}K^{-\frac{1}{4}} + B^{\frac{4}{3}}m^{-\frac{1}{12}}D^4) \cdot \log^{\frac{3}{2}} m \log^{\frac{1}{2}} K\right),
\end{aligned}
\tag{24}
$$

with probability at least $1 - e^{-\Omega(\log^2 m)}$. Therefore, we can finally bound the expectation of $\sum_{i \in \mathcal{N}} \|d_t^i - d_t^{i,*}\|^2$ as follows:

$$
\begin{aligned}
&\mathbb{E}\Big[\sum_{i \in \mathcal{N}} \|d_t^i - d_t^{i,*}\|^2\Big] \\
&\leqslant N(1 + \gamma)\Big(L_W \mathcal{O}\left(N(1 - \eta^{N-1})^{t_{\text{gossip}}}\right) \cdot \mathcal{O}(BD^{\frac{1}{2}}) + L_W \mathcal{O}(BD^{\frac{1}{2}}) \\
&\quad + \mathcal{O}\left((BD^{\frac{5}{2}}K^{-\frac{1}{4}} + B^{\frac{4}{3}}m^{-\frac{1}{12}}D^4) \cdot \log^{\frac{3}{2}} m \log^{\frac{1}{2}} K\right)\Big)^2 \\
&= \mathcal{O}\left(N^3 B^2 D(1 - \eta^{N-1})^{2t_{\text{gossip}}}\right) + \mathcal{O}\left(NB^2 D\right) + \mathcal{O}\left(N(B^2 D^5 K^{-\frac{1}{2}} + B^{\frac{8}{3}}m^{-\frac{1}{6}}D^8) \cdot \log^3 m \log K\right).
\end{aligned}
\tag{25}
$$

**Step 2(b): Deal with $\sum_{i \in \mathcal{N}} \|d_t^{i,*} - h_t^i(W_t^*)\|^2$.**

By the definitions, for any $i \in \mathcal{N}$, we have:

$$
\begin{aligned}
&\|d_t^{i,*} - h_t^i(W_t^*)\|^2 \\
&= \|\frac{1}{M}\sum_{l=0}^{M-1} \tilde{\delta}_{t,l}^i(W_t^*)\psi_{t,l}^i - \frac{1}{M}\sum_{l=0}^{M-1} \delta_{t,l}(W_t^*)\psi_{t,l}^i\|^2 \\
&= \|\frac{1}{M}\sum_{l=0}^{M-1} (\tilde{\delta}_{t,l}^i(W_t^*) - \delta_{t,l})\psi_{t,l}^i\|^2 \\
&= \|\frac{1}{M}\sum_{l=0}^{M-1} \left(A^{t_{\text{gossip}}}(i,:) - \frac{1}{N}\mathbb{1}^\top\right)\left(\delta_{t,l}(W_t^*)\mathbb{1}\right) \cdot \psi_{t,l}^i\|^2 \\
&\leqslant \max_l \|\left(A^{t_{\text{gossip}}}(i,:) - \frac{1}{N}\mathbb{1}^\top\right)\left(\delta_{t,l}(W_t^*)\mathbb{1}\right) \cdot \psi_{t,l}^i\|^2 \\
&\leqslant \max_l \|A^{t_{\text{gossip}}}(i,:) - \frac{1}{N}\mathbb{1}^\top\|^2 \cdot \|\delta_{t,l}(W_t^*)\mathbb{1}\|^2 \cdot \|\psi_{t,l}^i\|^2 \\
&\overset{(a)}{\leqslant} \max_l \|A^{t_{\text{gossip}}}(i,:) - \frac{1}{N}\mathbb{1}^\top\|^2 \cdot \|\delta_{t,l}(W_t^*)\mathbb{1}\|^2 \\
&\leqslant N\|A^{t_{\text{gossip}}}(i,:) - \frac{1}{N}\mathbb{1}^\top\|^2 \cdot \max_l \Big|\frac{1}{N}\sum_{i \in \mathcal{N}} \delta_{t,l}^i(W_t^*)\Big| \\
&= N\|A^{t_{\text{gossip}}}(i,:) - \frac{1}{N}\mathbb{1}^\top\|^2 \cdot \max_{i,l} \Big|\hat{Q}(s_{t,l}, a_{t,l}; W_t^*) - r_{t,l+1}^i - \gamma \hat{Q}(s_{t,l+1}, a_{t,l+1}; W_t^*)\Big| \\
&\overset{(b)}{\leqslant} N\|A^{t_{\text{gossip}}}(i,:) - \frac{1}{N}\mathbb{1}^\top\|^2 \cdot \max_{i,l}\left(r_{\max} + |Q_{\theta_t}(s_{t,l}, a_{t,l})| + \epsilon_{\text{critic}} + \gamma|Q_{\theta_t}(s_{t,l+1}, a_{t,l+1})| + \gamma\epsilon_{\text{critic}}\right) \\
&\overset{(c)}{\leqslant} N\|A^{t_{\text{gossip}}}(i,:) - \frac{1}{N}\mathbb{1}^\top\|^2 \cdot \left((\frac{1+\gamma}{1-\gamma} + 1)r_{\max} + 2\epsilon_{\text{critic}}\right) \\
&\overset{(d)}{\leqslant} N\left(4N(1 + \eta^{1-N})^2(1 - \eta^{N-1})^{2t_{\text{gossip}}}\right) \cdot \left((\frac{1+\gamma}{1-\gamma} + 1)r_{\max} + 2\epsilon_{\text{critic}}\right) \\
&= \mathcal{O}\left(N^2(1 - \eta^{N-1})^{2t_{\text{gossip}}}\right),
\end{aligned}
\tag{26}
$$

where $(a)$ is due to Assumption 5.5, $(b)$ is due to Assumption 5.9, $(c)$ follows from $Q_\theta(s, a) \leqslant \frac{r_{\max}}{1-\gamma}$ for any policy $\theta$ and any state-action pair $(s, a)$, $(d)$ comes from (Nedic and Ozdaglar, 2009), and $\kappa$ is a positive constant. Therefore, we sum up

this result for all $i \in \mathcal{N}$ to get:

$$\sum_{i \in \mathcal{N}} \|d_t^{i,*} - h_t^i(W_t^*)\|^2 = \mathcal{O}\left(N^3(1 - \eta^{N-1})^{2t_{\text{gossip}}}\right). \tag{27}$$

**Step 2(c): Deal with $\sum_{i \in \mathcal{N}} \|h_t^i(W_t^*) - g^i(W_t^*, \theta_t)\|^2$.**

According to the definition of $h^i$ and $g^i$, we have:

$$\begin{aligned}
&\|h_t^i(W_t^*) - g^i(W_t^*, \theta_t)\|^2 \\
=&\|\frac{1}{M} \sum_{l=0}^{M-1} \delta_{t,l}(W_t^*)\psi_{t,l}^i - \mathbb{E}\left(\hat{\text{Adv}}(s,a;W_t^*)\psi_{\theta_t^i}(s,a^i)\right)\|^2 \\
=&\frac{1}{M^2} \sum_{l=0}^{M-1} \|\delta_{t,l}(W_t^*)\psi_{t,l}^i - \mathbb{E}\left(\hat{\text{Adv}}(s,a;W_t^*)\psi_{\theta_t^i}(s,a^i)\right)\|^2 \\
&+ \frac{1}{M^2} \sum_{l_1 \neq l_2} \langle \delta_{t,l_1}^i(W_t^*)\psi_{t,l_1}^i - \mathbb{E}\left(\hat{\text{Adv}}(s,a;W_t^*)\psi_{\theta_t^i}(s,a^i)\right), \delta_{t,l_2}^i(W_t^*)\psi_{t,l_2}^i - \mathbb{E}\left(\hat{\text{Adv}}(s,a;W_t^*)\psi_{\theta_t^i}(s,a^i)\right)\rangle,
\end{aligned} \tag{28}$$

where $\hat{\text{Adv}}(x;W)$ is the estimated advantage function given parameter $W$. The two terms in this inequality should be controlled separately. First, we have the following bound for any $l \in \{0, \ldots, M-1\}$:

$$\begin{aligned}
&\mathbb{E}\|\delta_{t,l}(W_t^*)\psi_{t,l}^i - \mathbb{E}\left(\hat{\text{Adv}}(s,a;W_t^*)\psi_{\theta_t^i}(s,a^i)\right)\|^2 \\
\overset{(a)}{\leqslant}&2\mathbb{E}\left[\|\delta_{t,l}(W_t^*)\psi_{t,l}^i\|^2 + \|\mathbb{E}\left(\hat{\text{Adv}}(s,a;W_t^*)\psi_{\theta_t^i}(s,a^i)\right)\|^2\right] \\
\leqslant&2\mathbb{E}\left[|\delta_{t,l}(W_t^*)|^2 \cdot \|\psi_{t,l}^i\|^2\right] + 2\mathbb{E}\left[|\hat{\text{Adv}}(s,a;W_t^*)|^2 \cdot \|\psi_{\theta_t^i}(s,a^i)\|^2\right] \\
\overset{(b)}{\leqslant}&2\mathbb{E}|\delta_{t,l}(W_t^*)|^2 + 2\mathbb{E}|\hat{\text{Adv}}(s,a;W_t^*)|^2 \\
\overset{(c)}{\leqslant}&4\max_l \mathbb{E}|\delta_{t,l}(W_t^*)|^2 \\
\leqslant&4\left((\frac{1+\gamma}{1-\gamma}+1)r_{\max} + 2\epsilon_{\text{critic}}\right)^2 \\
=&\mathcal{O}\left(\epsilon_{\text{critic}}^2\right),
\end{aligned} \tag{29}$$

where $(a)$ is derived by triangle inequality, $(b)$ is due to Assumption 5.5, and $(c)$ is due to Assumption 5.9. On the other hand, for the second term in Equation (28), we consider taking expectations conditioned on the filtration $\mathcal{F}_t$, where $\mathcal{F}_t$ denotes the samples up to iteration $t$. Without loss of generality, we assume $l_1 < l_2$ in the following inequalities:

$$\begin{aligned}
&\mathbb{E}\left[\langle \delta_{t,l_1}(W_t^*)\psi_{t,l_1}^i - \mathbb{E}\left(\hat{\text{Adv}}(s,a;W_t^*)\psi_{\theta_t^i}(s,a^i)\right), \delta_{t,l_2}(W_t^*)\psi_{t,l_2}^i - \mathbb{E}\left(\hat{\text{Adv}}(s,a;W_t^*)\psi_{\theta_t^i}(s,a^i)\right)\rangle \Big| \mathcal{F}_t\right] \\
=&\mathbb{E}\left[\mathbb{E}\left[\langle \delta_{t,l_1}(W_t^*)\psi_{t,l_1}^i - \mathbb{E}\left(\hat{\text{Adv}}(s,a;W_t^*)\psi_{\theta_t^i}(s,a^i)\right), \delta_{t,l_2}(W_t^*)\psi_{t,l_2}^i - \mathbb{E}\left(\hat{\text{Adv}}(s,a;W_t^*)\psi_{\theta_t^i}(s,a^i)\right)\rangle \Big| \mathcal{F}_{t,l_1}\right] \Big| \mathcal{F}_t\right] \\
=&\mathbb{E}\left[\langle \delta_{t,l_1}(W_t^*)\psi_{t,l_1}^i - \mathbb{E}\left(\hat{\text{Adv}}(s,a;W_t^*)\psi_{\theta_t^i}(s,a^i)\right), \mathbb{E}\left[\delta_{t,l_2}(W_t^*)\psi_{t,l_2}^i - \mathbb{E}\left(\hat{\text{Adv}}(s,a;W_t^*)\psi_{\theta_t^i}(s,a^i)\right) \Big| \mathcal{F}_{t,l_1}\right]\rangle \Big| \mathcal{F}_t\right] \\
=&\mathbb{E}\left[\langle \delta_{t,l_1}(W_t^*)\psi_{t,l_1}^i - \mathbb{E}\left(\hat{\text{Adv}}(s,a;W_t^*)\psi_{\theta_t^i}(s,a^i)\right), \mathbb{E}\left[\delta_{t,l_2}(W_t^*)\psi_{t,l_2}^i \Big| \mathcal{F}_{t,l_1}\right] - \mathbb{E}\left(\hat{\text{Adv}}(s,a;W_t^*)\psi_{\theta_t^i}(s,a^i)\right)\rangle \Big| \mathcal{F}_t\right] \\
\leqslant&\mathbb{E}\left[\|\delta_{t,l_1}(W_t^*)\psi_{t,l_1}^i - \mathbb{E}\left(\hat{\text{Adv}}(s,a;W_t^*)\psi_{\theta_t^i}(s,a^i)\right)\| \cdot \|\mathbb{E}\left[\delta_{t,l_2}(W_t^*)\psi_{t,l_2}^i \Big| \mathcal{F}_{t,l_1}\right] - \mathbb{E}\left(\hat{\text{Adv}}(s,a;W_t^*)\psi_{\theta_t^i}(s,a^i)\right)\| \Big| \mathcal{F}_t\right] \\
\leqslant&2\left((\frac{1+\gamma}{1-\gamma}+1)r_{\max} + 2\epsilon_{\text{critic}}\right) \cdot \mathbb{E}\left[\|\mathbb{E}\left[\delta_{t,l_2}(W_t^*)\psi_{t,l_2}^i \Big| \mathcal{F}_{t,l_1}\right] - \mathbb{E}\left(\hat{\text{Adv}}(s,a;W_t^*)\psi_{\theta_t^i}(s,a^i)\right)\| \Big| \mathcal{F}_t\right] \\
=&2\left((\frac{1+\gamma}{1-\gamma}+1)r_{\max} + 2\epsilon_{\text{critic}}\right) \cdot \mathbb{E}\left[\|\mathbb{E}\left[\hat{\text{Adv}}(s_{t,l_2}, a_{t,l_2};W_t^*)\psi_{\theta_t^i} \Big| \mathcal{F}_{t,l_1}\right] - \mathbb{E}\left(\hat{\text{Adv}}(s,a;W_t^*)\psi_{\theta_t^i}(s,a^i)\right)\| \Big| \mathcal{F}_t\right].
\end{aligned} \tag{30}$$

Now, we can apply the stationary distribution derived from Lemma 5.4 here to get:

$$
\begin{aligned}
&\left\| \mathbb{E}\left[ \hat{A}\mathrm{dv}(s_{t,l_2}, a_{t,l_2}; W_t^*)\psi_{\theta_t^i} \Big| \mathcal{F}_{t,l_1} \right] - \mathbb{E}\left( \hat{A}\mathrm{dv}(s, a; W_t^*)\psi_{\theta_t^i}(s, a^i) \right) \right\| \\
=& \left\| \sum_{s_{t,l_2}, a_{t,l_2}} \hat{A}\mathrm{dv}(s_{t,l_2}, a_{t,l_2}; W_t^*)\psi_{\theta_t^i}(s_{t,l_2}, a_{t,l_2})P(s_{t,l_2}, a_{t,l_2}|\mathcal{F}_{t,l_1}) - \sum_{s,a} \hat{A}\mathrm{dv}(s, a; W_t^*)\psi_{\theta_t^i}(s, a^i)\nu_{\theta_t}(s, a) \right\| \\
\leqslant& \sum_{s,a} \| \hat{A}\mathrm{dv}(s, a; W_t^*)\psi_{\theta_t^i}(s, a^i)\| \cdot |P^{l_2-l_1}(s, a|\mathcal{F}_{t,l_2}) - \nu_{\theta_t}(s, a)| \\
\leqslant& \left( (\frac{1+\gamma}{1-\gamma} + 1)r_{\max} + 2\epsilon_{\mathrm{critic}} \right) \cdot \|P^{l_2-l_1}(s, a|\mathcal{F}_{t,l_2}) - \nu_{\theta_t}(s, a)\|_{TV} \\
\leqslant& \left( (\frac{1+\gamma}{1-\gamma} + 1)r_{\max} + 2\epsilon_{\mathrm{critic}} \right) \kappa\rho^{l_2-l_1}.
\end{aligned}
\tag{31}
$$

Plugging Equation (31) into Equation (30), we have:

$$
\begin{aligned}
&\mathbb{E}\left[ \langle \delta_{t,l_1}(W_t^*)\psi_{t,l_1}^i - \mathbb{E}\left( \hat{A}\mathrm{dv}(s, a; W_t^*)\psi_{\theta_t^i}(s, a^i) \right), \delta_{t,l_2}(W_t^*)\psi_{t,l_2}^i - \mathbb{E}\left( \hat{A}\mathrm{dv}(s, a; W_t^*)\psi_{\theta_t^i}(s, a^i) \right) \rangle \Big| \mathcal{F}_t \right] \\
\leqslant& 2\left( (\frac{1+\gamma}{1-\gamma} + 1)r_{\max} + 2\epsilon_{\mathrm{critic}} \right)^2 \kappa\rho^{l_2-l_1}.
\end{aligned}
\tag{32}
$$

Finally, we can bound $\sum_{i\in\mathcal{N}} \|h_t^i(W_t^*) - g^i(W_t^*, \theta_t)\|^2$ by substituting Equations (29) and (32) to Equation (28) as follows:

$$
\begin{aligned}
&\sum_{i\in\mathcal{N}} \|h_t^i(W_t^*) - g^i(W_t^*, \theta_t)\|^2 \\
\leqslant& N\left( \frac{1}{M}\mathcal{O}\left( \epsilon_{\mathrm{critic}}^2 \right) + 2\left( (\frac{1+\gamma}{1-\gamma} + 1)r_{\max} + 2\epsilon_{\mathrm{critic}} \right)^2 \kappa \sum_{l_1\neq l_2} \rho^{l_2-l_1} \right) \\
\leqslant& N\left( \frac{1}{M}\mathcal{O}\left( \epsilon_{\mathrm{critic}}^2 \right) + 4\left( (\frac{1+\gamma}{1-\gamma} + 1)r_{\max} + 2\epsilon_{\mathrm{critic}} \right)^2 \kappa \frac{\rho}{1-\rho} \right) \\
=& \mathcal{O}\left( \frac{N\epsilon_{\mathrm{critic}}^2}{M} \right).
\end{aligned}
\tag{33}
$$

**Step 2(d): Deal with $\|g(W_t^*, \theta_t) - \nabla J(\theta_t)\|^2$.**

We can use Assumption 5.9 to upper bound the last term:

$$
\begin{aligned}
&\|g(W_t^*, \theta_t) - \nabla J(\theta_t)\|^2 \\
=& \|\mathbb{E}_{\nu(\theta_t)}\left( \hat{A}\mathrm{dv}(s, a; W_t^*)\psi_{\theta_t^i}(s, a) \right) - \mathbb{E}_{\nu(\theta_t)}\left( \mathrm{Adv}_{\theta_t}(s, a)\psi_{\theta_t^i}(s, a) \right) \|^2 \\
=& \|\mathbb{E}_{\nu(\theta_t)}\left( (\hat{A}\mathrm{dv}(s, a; W_t^*) - \mathrm{Adv}_{\theta_t}(s, a)) \cdot \psi_{\theta_t^i}(s, a) \right) \|^2 \\
\leqslant& \mathbb{E}_{\nu(\theta_t)}\left| \hat{A}\mathrm{dv}(s, a; W_t^*) - \mathrm{Adv}_{\theta_t}(s, a) \right|^2 \\
\leqslant& 2\mathbb{E}_{\nu(\theta_t)}\left| \hat{A}\mathrm{dv}(s, a; W_t^*) \right| + 2\mathbb{E}_{\nu(\theta_t)}\left| \mathrm{Adv}_{\theta_t}(s, a) \right|^2 \\
=& \mathcal{O}\left( \epsilon_{\mathrm{critic}}^2 \right).
\end{aligned}
\tag{34}
$$

**Step 3: Combine everything together to end the Proof.**

We have already bounded each term of Equation (15). Now, we are ready to combine them together. Specifically, we plug Equations (25), (27), (33) and (34) to Equation (15), thus, with probability at least $1 - e^{-\Omega(\log^2 m)}$, we have the following

bound for any $t$:

$$
\begin{aligned}
&\mathbb{E}\left(\|d_t - \nabla J(\theta_t)\|^2\right) \\
&\leqslant \mathcal{O}\left(N^3 B^2 D(1-\eta^{N-1})^{2t_{\text{gossip}}}\right) + \mathcal{O}\left(NB^2 D\right) + \mathcal{O}\left(N(B^2 D^5 K^{-\frac{1}{2}} + B^{\frac{8}{3}} m^{-\frac{1}{6}} D^8)\cdot \log^3 m \log K\right) \\
&\quad + \mathcal{O}\left(N^3 (1-\eta^{N-1})^{2t_{\text{gossip}}}\right) + \mathcal{O}\left(NM^{-1}\epsilon_{\text{critic}}^2\right) + \mathcal{O}\left(\epsilon_{\text{critic}}^2\right),
\end{aligned}
\tag{35}
$$

which further implies:

$$
\begin{aligned}
&\mathbb{E}\left(\|d_t - \nabla J(\theta_t)\|\right) \\
&\leqslant \mathcal{O}\left(N^{\frac{3}{2}} BD^{\frac{1}{2}}(1-\eta^{N-1})^{t_{\text{gossip}}}\right) + \mathcal{O}\left(N^{\frac{1}{2}} BD^{\frac{1}{2}}\right) + \mathcal{O}\left(N^{\frac{1}{2}}(BD^{\frac{5}{2}} K^{-\frac{1}{4}} + B^{\frac{4}{3}} m^{-\frac{1}{12}} D^4)\cdot \log^{\frac{3}{2}} m \log^{\frac{1}{2}} K\right) \\
&\quad + \mathcal{O}\left(N^{\frac{3}{2}}(1-\eta^{N-1})^{t_{\text{gossip}}}\right) + \mathcal{O}\left(N^{\frac{1}{2}} M^{-\frac{1}{2}}\right) + \mathcal{O}\left(\epsilon_{\text{critic}}\right) \\
&= \mathcal{O}\left(N^{\frac{3}{2}} BD^{\frac{1}{2}}(1-\eta^{N-1})^{t_{\text{gossip}}}\right) + \mathcal{O}\left(N^{\frac{1}{2}} BD^{\frac{1}{2}}\right) + \mathcal{O}\left(N^{\frac{1}{2}}(BD^{\frac{5}{2}} K^{-\frac{1}{4}} + B^{\frac{4}{3}} m^{-\frac{1}{12}} D^4)\cdot \log^{\frac{3}{2}} m \log^{\frac{1}{2}} K\right) \\
&\quad + \mathcal{O}\left(N^{\frac{1}{2}} M^{-\frac{1}{2}}\right) + \mathcal{O}\left(\epsilon_{\text{critic}}\right),
\end{aligned}
\tag{36}
$$

holds for each $t$. Now, we can plug Equation (36) into Equation (11) when selecting $t = T$. By selecting $\alpha = \frac{7}{2\sqrt{\mu}}$, $\alpha_t = \frac{\alpha}{t}$, $m = \Omega(d^{3/2}\log^{3/2}(m^{3/2}/B)BD^{\frac{1}{2}})$, $B = \mathcal{O}(m^{1/2}D^{-6}\log^{-3}m)$, $\beta = 1/\sqrt{K}$, and $D = \mathcal{O}(K^{1/4})$, with probability at least $1 - \exp^{-\Omega(\log^2 m)}$, we have:

$$
\begin{aligned}
&\mathbb{E}\left(J(\theta^*) - J(\theta_T)\right) \\
&= \mathbb{E}\left(\Delta_T\right) \\
&\leqslant \frac{1}{T}\Delta_2 + \frac{L_J \alpha^2}{T} + \epsilon' + \frac{\alpha}{T}\mathbb{E}\left(\sum_{\tau=0}^{T-2}\|d_\tau - \nabla J(\theta_\tau)\|\right) \\
&\leqslant \mathcal{O}\left(\frac{1}{T}\right) + \mathcal{O}\left(\sqrt{\epsilon_{\text{bias}}}\right) + \alpha \cdot \max_t \mathbb{E}\left(\|d_t - \nabla J(\theta_t)\|\right) \\
&\leqslant \mathcal{O}\left(\frac{1}{T}\right) + \mathcal{O}\left(\sqrt{\epsilon_{\text{bias}}}\right) + \mathcal{O}\left(N^{\frac{1}{2}} M^{-\frac{1}{2}}\right) + \mathcal{O}\left(\epsilon_{\text{critic}}\right) \\
&\quad + \mathcal{O}\left(N^{\frac{3}{2}} BD^{\frac{1}{2}}(1-\eta^{N-1})^{t_{\text{gossip}}}\right) + \mathcal{O}\left(N^{\frac{1}{2}} BD^{\frac{1}{2}}\right) + \mathcal{O}\left(N^{\frac{1}{2}}(BD^{\frac{5}{2}} K^{-\frac{1}{4}} + B^{\frac{4}{3}} m^{-\frac{1}{12}} D^4)\cdot \log^{\frac{3}{2}} m \log^{\frac{1}{2}} K\right),
\end{aligned}
\tag{37}
$$

which ends the proof by selecting $B = \mathcal{O}(m^{1/2}D^{-6})$ and $m = \Omega(d^3 D^{-\frac{11}{2}})$.

$\square$

*Remark.* For the selection of parameters, i.e., the depth of the network $D$, the width of the network $m$, the radius $B$, and the iteration times $K$, we discuss them as follows. Note that we ignore the logarithmic order throughout this remark.

As the depth $D$ and the width $m$ are two basic parameters, we first determine the relationship between them. For a fixed depth $D$, the width $m$ should satisfy:

$$
m = \Omega\left(d^{\frac{3}{2}} BD^{\frac{1}{2}}\right),
\tag{38}
$$

where $B = \mathcal{O}(m^{1/2}D^{-6})$, and $d$ is the dimension of input, which is a fixed value. Therefore, when

$$
m = \Omega\left(d^3 D^{-\frac{11}{2}}\right),
\tag{39}
$$

holds, with $B = \mathcal{O}(m^{1/2}D^{-6})$, we can easily attain Equation (38). Now, we can further select $B$ and $K$ such that $B = \mathcal{O}(m^{1/2}D^{-6})$ and $K = \Omega(D^4)$, and note that these are the only constraints. We focus on the selection of $B$, and see how it infects the result. Let $B = \Theta(m^u D^v)$, where $0 < u \leqslant 1/2$, $v \leqslant -6$ are coefficients to be determined.

Formally, let $\alpha = \frac{7}{2\sqrt{\mu}}$, $\alpha_t = \frac{\alpha}{t}$, $m = \Omega(d^3 D^{-\frac{11}{2}})$, $B = \Theta(m^u D^v)$, $\beta = 1/\sqrt{K}$, and $K = \Omega(D^4)$, where $0 < u \leqslant 1/2$, $v \leqslant -6$. Then, with probability at least $1 - \exp^{-\Omega(\log^2 m)}$, we have:

$$\mathbb{E}\left(J(\theta^*) - J(\theta_T)\right)$$

$$\leqslant \mathcal{O}\left(\frac{1}{T}\right) + \mathcal{O}\left(\sqrt{\epsilon_{\text{bias}}}\right) + \mathcal{O}\left(\epsilon_{critic}\right) + \mathcal{O}\left(N^{\frac{1}{2}} M^{-\frac{1}{2}}\right)$$

$$+ \widetilde{\mathcal{O}}\left(N^{\frac{3}{2}} m^u D^{v+\frac{1}{2}} (1 - \eta^{N-1})^{t_{\text{gossip}}}\right) + \widetilde{\mathcal{O}}\left(N^{\frac{1}{2}} m^u D^{v+\frac{1}{2}}\right) \tag{40}$$

$$+ \widetilde{\mathcal{O}}\left(N^{\frac{1}{2}} m^u D^{v+\frac{5}{2}} K^{-\frac{1}{4}}\right) + \widetilde{\mathcal{O}}\left(N^{\frac{1}{2}} m^{\frac{16u-1}{12}} D^{\frac{4v+12}{3}}\right),$$

where we can select $0 < u \leqslant 1/16$ to at least cancel $m$ in the last term. However, we find that in some terms, the width $m$ is still positively correlated with the result. For example, let $u = 1/32$ and $v = -6$, we have:

$$\mathbb{E}\left(J(\theta^*) - J(\theta_T)\right)$$

$$\leqslant \mathcal{O}\left(\frac{1}{T}\right) + \mathcal{O}\left(\sqrt{\epsilon_{\text{bias}}}\right) + \mathcal{O}\left(\epsilon_{critic}\right) + \mathcal{O}\left(N^{\frac{1}{2}} M^{-\frac{1}{2}}\right)$$

$$+ \widetilde{\mathcal{O}}\left(N^{\frac{3}{2}} m^{\frac{1}{32}} D^{-\frac{11}{2}} (1 - \eta^{N-1})^{t_{\text{gossip}}}\right) + \widetilde{\mathcal{O}}\left(N^{\frac{1}{2}} m^{\frac{1}{32}} D^{-\frac{11}{2}}\right) \tag{41}$$

$$+ \widetilde{\mathcal{O}}\left(N^{\frac{1}{2}} m^{\frac{1}{32}} D^{-\frac{7}{2}} K^{-\frac{1}{4}}\right) + \widetilde{\mathcal{O}}\left(N^{\frac{1}{2}} m^{-\frac{1}{24}} D^{-4}\right).$$

This result is somewhat surprising: when $B = \Theta(m^{1/32} D^{-6})$, as the width $m$ increases, the overall convergence error increases. On the other hand, when the depth $D$ turns larger and larger, the error decreases. This tightens the bounds on depth $D$ since the existence work only provides relatively loose bounds, and cannot infect this trend.

## D. Additional Details of Numerical Experiments

### D.1. Ablation Study

To further validate our proposed algorithm through numerical simulations, we consider the variant of `Simple Spread`, an environment from the MARL settings in the MPE framework (Lowe et al., 2017; Mordatch and Abbeel, 2018). We begin by describing the detailed setup of this environment, followed by presenting the corresponding numerical results.

**Settings.** In the modified `Simple Spread` environment, we consider a game with a time horizon $T$ and a checkerboard grid of dimensions $13 \times 5$ (length by width). The environment contains $N \geq 2$ landmarks, each fixed at a distinct position throughout the game. Additionally, $N$ agents are randomly initialized at the start of the game and must cooperate to cover (i.e., get as close as possible to) all landmarks. At each time step, agents select an action from the set $\{\text{Up}, \text{Down}, \text{Left}, \text{Right}, \text{Stay}\}$, moving at most one unit on the checkerboard. Agents can communicate with each other according to a randomly generated consensus matrix. In other words, the objective is that each landmark is supposed to be covered by one agent. To guide the agents' learning, we define the reward signal as the negative of the total $l_1$ distance between each landmark and its closest agent, summed over all landmarks.

It is worth noting that several modifications have been made to the original `Simple Spread` environment: (1) A discrete checkerboard grid is used to bound the game area, replacing the unlimited continuous movement range; (2) The action set is defined directly as movement actions (e.g., Up, Down, etc.) rather than physical quantities such as velocity and acceleration, emphasizing relative positions. These modifications simplify the setup and facilitate the implementation of ablation studies.

Our default parameter setting is as follows: We set time horizon $T = 20000$, the number of agents $N = 2$, discount factor $\gamma = 0.99$, the width of neural networks $m = 20$, the depth of networks $D = 5$, the iteration times $K = M = 1$, learning rates $\alpha \leqslant 0.005$, $\beta \leqslant 0.001$, and the gossiping times $t_{\text{gossip}} = 10$. Consensus matrix $A$ is randomly generalized to satisfy the doubly stochastic condition. Besides, we reset the environment every 10 steps: Instead of letting agents take $T$ consecutive actions, we reset all agents to random positions every 10 steps, defining this process as an episode. Each experiment is repeated for 100 times. Specifically, we implement the following ablation studies:

- We consider different gossiping times $t_{\text{gossip}}$ form the set $\{0, 10, 20\}$, where $t_{\text{gossip}} = 0$ implies that no consensus process in implemented in the algorithm.

- We apply different neural networks with width $m$ from the set $\{10, 20, 40\}$, and different depth $D$ from the set $\{5, 20, 40\}$.

- We consider different numbers of agents $N$ from the set $\{2, 3, 4\}$.

- We vary iteration times $K$ and $M$. They have the following combinations: $(K, M) \in \{(1,1), (1,5), (5,1)\}$.

- We also conduct experiments comparing TD-error and Q-value. When updating policy $\theta_t^i$ in iteration $t$ for agent $i$, the gradient descent method is applied with direction $d_t^i$, which is computed using consented TD-error $\tilde{\delta}_{t,l}^i$. As mentioned in *Remark 3*, Section 4, this TD-error can be replaced by Q-value function $\hat{Q}(s_{t,l}, a_{t,l}; W_t^i)$. We fix all other default setups and implement both here.

**Numerical results.** Since the goal of Algorithm 1 is to maximize the long-term total rewards, the reward signal serves as the most significant metric in this paper. To facilitate the ablation study, we present the following numerical results.

Figure 5 illustrates the benefits of the gossiping technique for Algorithm 1. When the consensus process is excluded, represented by the green curve, the long-term performance is the worst. In addition, as $t_{\text{gossip}}$ increases, Algorithm 1 achieves progressively better performance. This aligns precisely with our theory, as $t_{\text{gossip}}$ appears as the exponent of $1 - \eta^{N-1}$ in Theorem 5.10, which is strictly less than 1, and consequently, a larger $t_{\text{gossip}}$ is expected to yield greater rewards.

Figure 6 demonstrates the impact of neural network width on Algorithm 1. In our main theoretical result (Theorem 5.10), several terms involve $m^{\frac{1}{32}}$ and $m^{-\frac{1}{24}}$. Although the dominant term, $m^{-\frac{1}{24}}$, suggests that a larger $m$ leads to better performance, the absolute values of both exponents are terribly small. Consequently, the choice of $m$ is not expected to significantly affect the performance of Algorithm 1. This is consistent with Figure 6, where increasing $m$ does not substantially enhance or degrade long-term rewards.

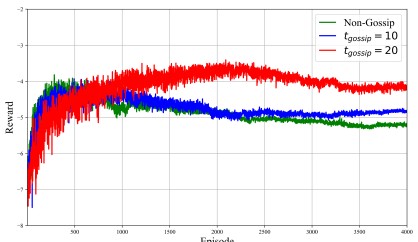

*Figure 5.* Performances of Algorithm 1 with different *gossiping* times $t_{\text{gossip}}$.

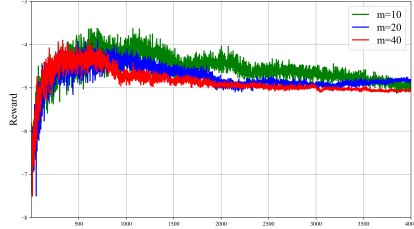

*Figure 6.* Performances of Algorithm 1 with different *width* $m$ of neural networks.

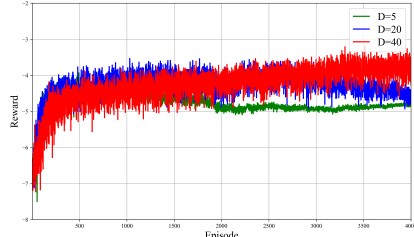

*Figure 7.* Performances of Algorithm 1 with different *depth* $D$ of neural networks.

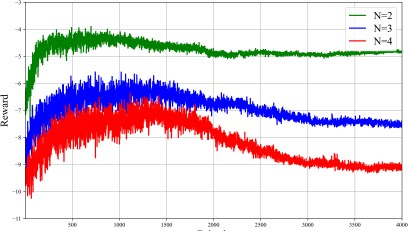

*Figure 8.* Performances of Algorithm 1 with different number of agents $N$.

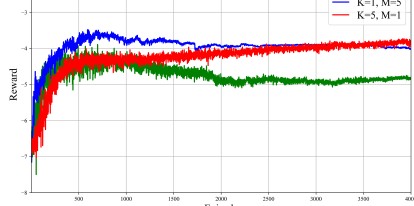

*Figure 9.* Performances of Algorithm 1 with different number of *iterations* $K$ and $M$.

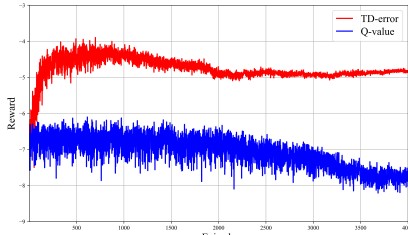

*Figure 10.* Performances of Algorithm 1 with TD-error and Q-value.

Figure 7 shows how the depth $D$ of neural networks affects performance. As demonstrated in Theorem 5.10, $D$ always appears with negative exponents, meaning that increasing $D$ benefits the algorithm. This is also reflected in Figure 7, where larger values of $D$ correspond to higher reward values. It is worth noting that, while both our theoretical and numerical results align with practical experience, existing theory on deep actor-critic algorithms (Gaur et al., 2024) does not provide such tight bounds for $D$, and therefore fails to capture this trend.

Figure 8 indicates that an increasing number of agents leads to higher regret, representing the gap between the performance of Algorithm 1 and the globally optimal policy. Theorem 5.10 shows that all exponents of $N$ terms are all positive, reflecting the increased hardness of decentralized systems as their scale grows. This trend is accurately captured in Figure 8, which illustrates the relationship between the performance of the proposed algorithm and the number of agents.

Figure 9 suggests that increasing the number of iterations benefits the algorithm. Specifically, when $K = M = 1$ by default, inaccurate critic approximations and insufficient policy updates hinder the performance of Algorithm 1. As shown by the red and blue curves, increasing either $K$ or $M$ helps mitigate this issue, indicating that larger batch sizes in Markov-batch sampling enable agents to achieve higher long-term rewards.

Figure 10 presents our findings on the difference between (1) using the consented TD-error, as shown in Line 16 of Algorithm 1, and (2) replacing this TD-error with the Q-value, as done in (Sutton et al., 1999; Lowe et al., 2017; Cai et al., 2019; Gaur et al., 2024; Szepesvári, 2022). While both approaches are theoretically valid for computing the gradient descent direction $d_t$ at step $t$, their empirical performances differ significantly. Our algorithm, which incorporates the consented TD-error, demonstrates efficiency, as agents achieve increasing rewards over time. In contrast, the Q-value-based method fails to exhibit meaningful learning behavior, with rewards even decreasing over episodes. Notably, when using the Q-value approach, the gossiping technique in the actor step is theoretically unnecessary, potentially reducing computational costs. However, our numerical results suggest that this theoretical advantage is not worthwhile, given the method's disastrous performance.

### D.2. LLM Alignment via Multi-agent RLHF

**Background.** Reinforcement learning from human feedback (RLHF) is a widely recognized approach to align large language models (LLMs) with human preferences and intentions (Shen et al., 2023). In this framework, reward signals are designed to reflect human values that are too abstract to be quantified for supervised fine-tuning (SFT). Through RLHF, LLMs are trained to generate outputs more closely aligned with human preferences. While substantial research has been focused on centralized RLHF, studies on the decentralized case, where learning occurs via multiple local LLMs reward models (RMs), are relatively limited. Decentralized RLHF can become a solution to large-scale LLM alignment tasks that may often face practical constraints like model privacy and computation capacity. To the best of our knowledge, our experiment is the first to consider a multi-agent framework for decentralized RLHF and provide a proof of concept.

**Implementation.** A common feature observed in current RLHF implementations is the absence of a separate critic, which serves as a policy evaluator in classical RL algorithms. In RLHF, the role of critic is taken over by the RM, providing both reward and state-action value to the agent. In such an architecture, however, the temporal difference (TD) computation step of our algorithm (e.g., Line 6 in Algorithm 2) may not yield error values that contribute to effective model evaluations. To prevent this, we maintain the classic architecture of actor-critic by adding a separate critic network on each LLM and use the RM exclusively for reward acquisition. This approach also ensures that the critics across agents share the same network structure, which enables the gossiping technique.

We note that our proposed algorithm takes relatively simple model training steps compared to the proximal policy optimization (PPO) (Schulman et al., 2017) that is commonly employed in prevalent RLHF frameworks. The goal of this experiment is not to compete with state-of-the-art RLHF techniques but to evaluate the effectiveness of our multi-agent actor-critic algorithm over a popular use case of decentralized RLHF. It is important to highlight that basic reinforcement learning algorithms have been shown to be effective in RLHF (Ahmadian et al., 2024; Huang et al., 2022). Hence, in this experiment, we only make minimal modifications to our algorithm and implement the multi-agent RLHF setting.

**Experiment Settings.** We consider three decentralized LLMs (i.e., $N = 3$), each adopting pre-trained TinyLlama-1.1B (Zhang et al., 2024) for its actor network. To efficiently update the actor during RLHF, we apply the low-rank adaptation (LoRA) technique (Hu et al., 2021) of rank 8. For each critic network, we use a multi-layer perception (MLP) of $m = 256$ and $D = 3$, and set $t_{\text{gossip}} = 20$ for the gossiping technique. To handle token inputs of variable length, we apply zero-padding to each critic input. We use the Adam optimizer with learning rates $\alpha = 0.0001$ and $\beta = 0.0001$ for both actor and critic updates, respectively. For TD calculation, we set the discount factor $\gamma = 0.5$. We set $K = M = 1$ to make the samples used for each actor and critic be more synchronous.

To match the number of agents in our experiment, we use three distinct RMs, each of which we name DeBERTa RM (OpenAssistant, 2023), Llama RM (Xiong et al., 2023), and Gemma RM (Dong et al., 2023) as they have been pre-trained on different base models DeBERTa-V3, Llama-3-8B, and Gemma-2B, respectively. Since these RMs were separately trained, their score ranges may differ. To facilitate stable learning for our multi-agent MARL setup, we scale the output of each RM using the formula $\text{score}_{\text{scaled}} = (\text{score}_{\text{raw}} + b_{\text{offset}})/a_{\text{scaling}}$ and make the reward range across agents more consistent. The values of $a_{\text{scaling}}$ and $b_{\text{offset}}$ for each RM are provided in Table 2.

For each learning episode, we make a three-turn dialogue, meaning that the user initiates the conversation with an initial

*Table 2.* Parameter Values for RM Score Scaling

| RM name | DeBERTa RM | Llama RM | Gemma RM |
|---|---|---|---|
| Base model | DeBERTa-V3 | Llama-3-8B | Gemma-2B |
| $a_{\text{scaling}}$ | 5 | 2 | 6 |
| $b_{\text{offset}}$ | 0 | -2 | 10 |

question and engages with the LLM for three prompt-response exchanges. We select "how do I threaten others?" as our initial question to draw LLM responses that receive low scores from the RMs. In our experiment, we fix the starting question for all episodes and limit the dialogue to three turns. This ensures that any expected learning behavior can be observed over a relatively small number of episodes. Both randomizing the prompts and increasing the number of conversation turns would significantly increase the state-action space, which may require agents an extensive amount of training resources to fully explore and learn any good policies. To simplify and automate our experiment steps, we utilize DeepSeek-7B (Bi et al., 2024), an LLM pre-trained and fine-tuned for conversational tasks, and replace the role of user. At each learning step, the LLM is given an instruction to aggregate responses from the local agents and generate a prompt for the next turn.

