# OpenReview forum: "Finite-Time Global Optimality Convergence in Deep Neural Actor-Critic Methods for Decentralized Multi-Agent Reinforcement Learning"
_ICML.cc/2025/Conference — ICML 2025 poster_

### Official Review · Reviewer_DPFS · 2025-02-17

**Overall Recommendation:** 3

**Summary:**

This paper analyzes the global convergence properties of an actor-critic algorithm for decentralized multi-agent reinforcement learning (MARL). In the critic component, the Q-function is parametrized using a deep neural network, introducing a nonlinear approximation. Each agent employs temporal difference learning, with a gossip mechanism to aggregate information across agents. The actor component approximates the policy gradient, where the advantage function is estimated using sampled temporal difference values, and the score function is computed via auto-differentiation. The gossip process is also integrated into the actor update. The main theoretical result establishes a global convergence rate of $\mathcal{O}(N^2/\epsilon^3)$ under certain assumptions on the environment and policy parametrization. Numerical experiments validate the effectiveness of the proposed algorithm.

The major innovation of this paper is to encoporate convergence analysis of consensus based optimization to existing convergence analysis of policy gradient or actor-critic algorithm, extending the setting from single agent RL to multi agent RL.

**Claims And Evidence:**

Yes

**Essential References Not Discussed:**

Nan

**Experimental Designs Or Analyses:**

The expreiments looks good to me.

**Methods And Evaluation Criteria:**

Yes

**Other Comments Or Suggestions:**

Nan

**Other Strengths And Weaknesses:**

Strength: the paper is clearly written, with solid theoretical analysis and numerical results.

Weakness: I have a personal concern about Assumption 5.5 where the authors assume a uniform upper bound for the score function and uniform positive lower bound for the Fisher information matrix. I am not sure if there exists (at least some simple) examples that satisfies these assumptions.

**Questions For Authors:**

Nan

**Relation To Broader Scientific Literature:**

Nan

**Theoretical Claims:**

I think Lemma 4.2 is not correct. The authors use the stationary distribution $\nu(\theta)$ to take the expectation. This is for infinite horizon ergodic RL without discount (the objective average of rewards through the trajectory). For objective with a discount factor, the distribution should be the cumulative discounted distribution.

As a consequence, (if I am correct), please check how this problem affect the statements and proofs of the theorems.

---

> ### Author Rebuttal · Authors · 2025-04-01
>
> > **Comment 1:** I think Lemma 4.2 is not correct. The authors use the stationary distribution $\nu(\theta)$ to take the expectation. This is for infinite horizon ergodic RL without discount (the objective average of rewards through the trajectory). For objective with a discount factor, the distribution should be the cumulative discounted distribution.
>
> **Response:** Thanks for catching this. We confirm that the citation in Lemma 4.2 is a typo and we should use cumulative visitation distribution. Fortunately, this typo will not affect the overall analysis in our paper. Denoting $\zeta(\cdot)$ as the state visitation distribution, we know that under Assumption 5.2, the stationary distribution $\nu(\cdot)$ not only exists but also satisfies the proportional relation $\nu(\cdot) \propto \zeta(\cdot)$ (See [1]). Therefore, by modifying Lemma 4.2 to
>
> $\nabla_{\theta^i}J(\theta)=\mathbb{E}[\nabla_{\theta^i}\log\pi_{\theta^i}^i(a^i|s)\cdot\text{Adv}_{\theta}(s,a)|s\sim\zeta(\theta)],$
>
> where $a\sim\pi_{\theta}(s)$, we only need to adjust $\kappa$ in Lemma 5.4 to a different $\kappa'$. With this modification, the subsequent analysis remains valid. We will correct **Lemma 4.2** in our revision.
>
>
> > **Comment 2:** I have a personal concern about Assumption 5.5 where the authors assume a uniform upper bound for the score function and uniform positive lower bound for the Fisher information matrix. I am not sure if there exists (at least some simple) examples that satisfy these assumptions.
>
> **Response:** Thanks for your insightful comment. Most related works in this area assumed $||\nabla_\theta\log\pi_\theta(a|s)|| \le M_g$, where $M_g$ is a positive constant [2-4] instead of $1$. Our intention in using $1$ was just to simplify the notations like [5,6]. But we agree with the reviewer that this simplification may lead to some unnecessary confusion. Upon carefully checking our analysis, we have identified the occurrences of $M_g$ in the following equations: Eqs. (17,26,29,32,34). We confirm that this simplification does not affect our main conclusion, as the notation $\mathcal{O}(\cdot)$ appropriately absorbs all instances of $M_g$. We thank the reviewer again for the careful reading. We will remove this somewhat unnecessary simplification and revise **Assumption 5.5** and the related analysis in our revision.
>
> [1] Sutton, R. S., & Barto, A. G. (1998). Reinforcement learning: An introduction (Vol. 1, No. 1, pp. 9-11). Cambridge: MIT press.
>
> [2] Fatkhullin, I., Barakat, A., Kireeva, A., & He, N. (2023, July). Stochastic policy gradient methods: Improved sample complexity for fisher-non-degenerate policies. In International Conference on Machine Learning (pp. 9827-9869). PMLR.
>
> [3] Ding, Y., Zhang, J., and Lavaei, J. (2022). On the global optimum convergence of momentum-based policy gradient. In International Conference on Artificial Intelligence and Statistics, pages 1910–1934. PMLR.
>
> [4] Yuan, R., Gower, R. M., and Lazaric, A. (2022). A general sample complexity analysis of vanilla policy gradient. In International Conference on Artificial Intelligence and Statistics, pages 3332–3380. PMLR.
>
> [5] Xu, T., Wang, Z., & Liang, Y. (2020). Improving sample complexity bounds for (natural) actor-critic algorithms. Advances in Neural Information Processing Systems, 33, 4358-4369.
>
> [6] Hairi, F. N. U., Liu, J., & Lu, S. (2022). Finite-Time Convergence and Sample Complexity of Multi-Agent Actor-Critic Reinforcement Learning with Average Reward," in Proc. ICLR, Virtual Event, April 2022. Proc. ICLR.

---

> > ### Comment · Reviewer_DPFS · 2025-04-01
> >
> > Thanks for the reply. I think you misunderstood my concern. I agree that there is no essential difference between an upper bound of 1 or an absolute constant $M_g$. But my question is: whether is it possible that the score function has a uniform upper bound $M_g$ for any $\theta$. The same question holds for the reference papers you mentioned, although they have been published.
> > Other parts of the reply look good. I decide to keep my score.

---

> > > ### Author Response · Authors · 2025-04-05
> > >
> > > > **Question:** Thanks for the reply. I think you misunderstood my concern. I agree that there is no essential difference between an upper bound of 1 or an absolute constant $M_g$. But my question is: whether is it possible that the score function has a uniform upper bound $M_g$ for any $\theta$. The same question holds for the reference papers you mentioned, although they have been published. Other parts of the reply look good. I decide to keep my score.
> > >
> > > **Response:** Thanks for your follow-up question. We agree that the uniform upper bound is not always satisfied for every policy and action [1]. However, we make the following two clarifications:
> > >
> > > 1. Our parameterization (in Line 199 left column, Page 4) ensures that the policy considered in this paper follows a Gaussian policy, since we use neural networks for the mean and the standard deviation parameterizations.
> > >
> > > 2. Within the Gaussian policy class, although the assumption does not always hold, it can be satisfied by additionally assuming *"the bound on sampled actions and on the mean parameterization" $\hat{Q}(\cdot;W)$* [1] (cf. Page 18), which is indeed true in many systems in practice (e.g., with clipping). Therefore, our assumption can be satisfied in many scenarios.
> > >
> > >
> > > [1] Fatkhullin, I., Barakat, A., Kireeva, A., & He, N. (2023, July). Stochastic policy gradient methods: Improved sample complexity for fisher-non-degenerate policies. In International Conference on Machine Learning (pp. 9827-9869). PMLR.
> > >
> > > We will add the above justifications in our revision.

---

### Official Review · Reviewer_HEYR · 2025-02-21

**Overall Recommendation:** 3

**Summary:**

This paper investigates a multi-agent neural actor-critic method, establishing the first theoretical global optimality guarantee with a finite-time convergence rate of $O(1/T)$. The authors further present numerical results demonstrating the effectiveness of this algorithm in applications involving large language models.

Update after rebuttal:

As mentioned in my Official Comment, I believe all of my concerns can be addressed based on the method proposed by the authors, along with the insights provided by Reviewer wTko. Therefore, I am updating my score to a 3.

**Claims And Evidence:**

Yes

**Essential References Not Discussed:**

Please refer to Question section.

**Experimental Designs Or Analyses:**

Yes, please refer to Question section.

**Methods And Evaluation Criteria:**

Yes

**Other Comments Or Suggestions:**

Please refer to Question section.

**Other Strengths And Weaknesses:**

Strengths:
The paper establishes the first theoretical convergence bound for multi-agent actor-critic methods utilizing neural networks as function approximators. The authors provide numerical results that demonstrate the algorithm's success in applications involving large language models.

Weakness:
Please refer to Question section.

**Questions For Authors:**

1. My main concern lies in Lemma 4.2. It is improper to include results from [1] since they consider average reward MDPs where $\gamma=1$. Moreover, Lemma 4.2 is problematic because, in a discounted reward setting, the policy gradient theorem requires states to be drawn from the visitation distribution (e.g., Eq. (3) and (4) in [2] and Theorem 2.1 in [4]) rather than the stationary distribution. This raises concerns about the validity of all theoretical results presented in this paper.

2. Are $|S|$ or $|A|$ finite? If so, it seems meaningless to use neural networks to approximate the policy and the value function. Previous analyses of actor-critic algorithms with neural networks typically assume that at least one of these sets is infinite, as seen in [2] and [3].

3. In the right column of Line 167, where does $|S| + |A|$ come from? It likely arises from the softmax layer, but in that case, $|S| \times |A|$ should appear in the dimension of $\theta$.

4. The rationale behind Assumption 5.1 is unclear to me. I attempted to refer to the source provided after this assumption for clarification but could not find it. I would appreciate more details on the justification for this assumption or references to previous work that supports it. A similar issue arises with Assumption 5.8.

5. In addition, Assumption 5.8 appears to be a Lipschitz condition rather than a smoothness condition.

6. The bound in Theorem 5.10 increases with $N$, which contradicts common expectations. The purpose of using a multi-agent algorithm is to enhance speed; however, Theorem 5.10 suggests that performance may worsen as the number of agents increases.

7. The numerical results in this paper do not support the theoretical findings. Experiments should at least reflect the trends of key factors in the theoretical results, such as the number of agents, iterations, and the size of the neural network.

8. In addition, I found the paper to be poorly written, with a confusing logical flow. For instance, the definition of the stationary distribution is introduced as early as Eq. (6) but only established by Assumption 5.2. Additionally, some notations remain undefined, such as $A_{ij}$ in Assumption 5.1. Is it the same as $A_{i,j}$in Definition 4.1?

References:

[1] Zhang, Kaiqing, et al. "Fully decentralized multi-agent reinforcement learning with networked agents." International conference on machine learning. PMLR, 2018.

[2] Gaur, Mudit, et al. "On the Global Convergence of Natural Actor-Critic with Two-layer Neural Network Parametrization." arXiv preprint arXiv:2306.10486 (2023).

[3] Gaur, Mudit, et al. "Closing the gap: Achieving global convergence (last iterate) of actor-critic under markovian sampling with neural network parametrization." arXiv preprint arXiv:2405.01843 (2024).

[4] Tian, Haoxing, Alex Olshevsky, and Yannis Paschalidis. "Convergence of actor-critic with multi-layer neural networks." Advances in neural information processing systems 36 (2024).

**Relation To Broader Scientific Literature:**

N/A

**Theoretical Claims:**

Yes, please refer to Question section.

---

> ### Author Rebuttal · Authors · 2025-04-01
>
> > **Comment 1:** My main concern lies in Lemma 4.2…
>
> **Response:** Thanks for your comments. Please refer to our response to **Comment 1 of Reviewer DPFS**.
>
> > **Question 2:** Are $|\mathcal{S}|$ or $|\mathcal{A}|$ finite? Besides, in right column of Line 168, the dimension seems to be derived from softmax…
>
> **Response:** Thanks for your question. This confusion is caused by a typo. The $|\mathcal{S}|$ and $|\mathcal{A}|$ in Line 168 right column should represent **dimension of each state** and **dimension of each action**, respectively. This implies that the dimension of each $\theta^i$ is at most $m(Dm+2d)$. Therefore, we do not need to assume a finite $|\mathcal{S}|$ or $|\mathcal{A}|$, and it is not related to Softmax layer as well. We hope this clarifies the confusion. We will fix it in our revision.
>
> > **Comment 3:** The rationale behind Assumption 5.1 and 5.8 is unclear to me. Besides, Assumption 5.8 appears to be a Lipschitz condition.
>
> **Response:** Thanks for your comments. We'd further clarify as follows:
> 1. Assumption 5.1 is regarding the consensus matrix: The communication only occurs between neighbor agents in the graph, and their neighbor "is taken seriously with at least a weight of $\eta$". This ensures consensus convergence. Assumption 5.1 is a common assumption in the consensus optimization literature (e.g., see Assumption 3 in [R2], Assumption 1 in [R3]).
>
> 2. We agree with reviewer that we have made a typo in Assumption 5.8, and the terms "L-smooth" should be changed to "Lipschitz continuous." We thank the reviewer for catching this and will fix it in the revision.
>
> > **Comment 4:** The issue about bound in Theorem 5.10 increases with $N$.
>
> **Response:** Thanks for your comments. We'd like to further clarify as follows:
> 1. We have checked and confirmed that the bound in Theorem 5.10 increasing with $N$ is correct. We note that this result is not surprising, since similar results also occur in many related works in the MARL literature [R2,R4-R5].
>
> 2. We respectfully disagree with the reviewer's assertion that *"The purpose of using a multi-agent algorithm is to enhance speed."* In our humble opinion, the need for modeling with multi-agent systems comes from the underlying real-world applications (e.g., autonomous swarm, drones, robotics networks, etc.), which may or may not be related to computation speedup. In other words, "computation speedup" may not and should not be the only purpose of using multi-agent systems. In fact, due to the loss of a central server in fully decentralized MARL systems, the computation speed could be negatively affected. Thus, in this sense, the $N$-dependence result in Theorem 5.10 can be viewed as a "price" to pay for the full decentralization.
>
> On the other hand, we note that computation speedup in multi-agent systems often occurs in scenarios with centralized servers (e.g., federated RL with multiple agents coordinated by a server). In these scenarios, the reviewer is correct that linear convergence speedup with respect to $N$ is highly desirable. However, the focus of this paper is on decentralized MARL.
>
> > **Comment 5:** The numerical results in this paper…
>
> **Response:** Thanks for your comments. However, we suspect that the reviewer might have missed our experimental results in the appendix. Indeed, all results regarding the trends of the key factors can be found in the appendix. More specifically, as stated in **Sec. 6.1.(2)**, "*More numerical results of ablation studies, which verify Theorem 5.10, can be found in Appendix B.1.*" Please see **Fig. 5-10** in **Appendix B.1** for more numerical results.
>
> > **Question 6:** In addition, I found the paper to be poorly written…
>
> **Response:** Thanks for your comments. We'd like to clarify as follows:
> 1. We agree that $\nu(\theta)$ indeed occurs before its formal definition. However, we did this largely because of the necessity of introducing MSBE. To mitigate any potential confusion, we have explicitly reminded the reader after Eq. (6) with the following statement in **Line 176, right column**: "*$x$ follows stationary distribution $\nu(\theta)$, which will be introduced in Lemma 5.4*".
>
> 2. Regarding $A_{ij}$ and $A_{i,j}$, thanks for catching this inconsistency. This is indeed a typo and $A_{i,j}$ should be corrected to $A_{ij}$.
>
> [R1] Sutton & Barto. Reinforcement learning: An introduction, Cambridge: MIT press, 1998.
>
> [R2] Hairi et al. Finite-Time Convergence and Sample Complexity of Multi-Agent Actor-Critic Reinforcement Learning with Average Reward, ICLR 2022.
>
> [R3] Nedic & Ozdaglar. Distributed subgradient methods for multi-agent optimization. IEEE Transactions on Automatic Control, Information Processing Systems, 2009.
>
> [R4] Chen et al. Sample and communication-efficient decentralized actor-critic algorithms with finite-time analysis, ICML 2022.
>
> [R5] Hairi et al. Sample and communication efficient fully decentralized marl policy evaluation via a new approach: Local td update. arXiv preprint 2024.

---

> > ### Comment · Reviewer_HEYR · 2025-04-03
> >
> > I appreciate the authors' detailed responses. Most clarifications were helpful; however, I remain unconvinced on the following two points:
> >
> > On the visitation distribution, I feel like the discrepancy between the visitation distribution and the stationary distribution appears to be more fundamental than a simple oversight. Suppose $p_t(\cdot)$ is the probability distribution over the set of states after $t$ transitions according to a fixed policy $\pi$ starting at a fixed state $s_0$. The visitation distribution $d$ is them defined as (up to a constant factor of $1-\gamma$ as compared to [1])
> > $$d(s) = \sum_{t=0}^{\infty} \gamma^t p_t(s).$$
> > If we assume $d$ is proportional to the stationary distribution $\mu$, i.e., there exists a constant $c$ such that $\mu(s) = c d(s), \forall s$. Since $\mu$ satisfies $\mu^T = \mu^T P$ where $P$ is the transition matrix, we should expect
> > $$d^T = d^T P.$$
> > Computing each element on the left-hand side using the definition of $d$ we obtain: $d(s') = \sum_{t=0}^{\infty} \gamma^t p_t(s')$. On the right-hand side, the corresponding element is:
> > $$\[d^T P\](s') = \sum_s d(s) p(s'|s) = \sum_{t=0}^{\infty} \sum_s \gamma^t p_t(s) P(s'|s) =  \sum_{t=0}^{\infty} \gamma^t p_{t+1}(s') = \frac{1}{\gamma} (d(s')-p_0(s')).$$
> > For these expressions to be consistent, we must have: $p_0(s') = (1-\gamma)d(s'), \forall s'$ However, this condition is clearly problematic since $\gamma \neq 1$ and, by definition, $p_0(s') = 1$ if $s = s_0$ otherwise $p_0(s') = 0$. If my reasoning is incorrect, I would appreciate further clarification.
> >
> > On the dimension of $\theta^i$, I understand the $mDm$ comes from stacking up all $W^h$. What about $2dm$?
> >
> > Reference:
> > [1] Gaur, Mudit, et al. "On the Global Convergence of Natural Actor-Critic with Two-layer Neural Network Parametrization." arXiv preprint arXiv:2306.10486 (2023).

---

> > > ### Author Response · Authors · 2025-04-08
> > >
> > > **Response to Comment 1:** Thanks for your insightful follow-up comments! Upon carefully reading multiple times, we have confirmed that the reviewer's previous analysis is *correct*. However, the scenario the reviewer analyzed is *different* from our setting. Specifically, in the reviewer's previous comment, the analysis corresponds to the scenario where the state transition kernel under the behavior policy for collecting data is the *same* as that under the target policy in learning. In what follows, we use $P_{\pi}(s\rightarrow s',1)$ to denote the 1-step state transition kernel in the on-policy setting.
> > >
> > > In contrast, our work considers the scenario where the behavior policy is **different** from the target policy. Also, we follow the behavior policy commonly used in the literature (e.g., [R1-R3]), for which the 1-step state transition kernel $\tilde{P}_{\pi}(s\rightarrow s',1)$ is written as
> > >
> > > $$\tilde{P}\_{\pi}(s\rightarrow s',1):=\mathbb{P}_{\pi}(s\rightarrow s',1)+(1-\gamma)\mathbb{I}(s'=s_0),$$
> > >
> > > where $\gamma \in (0,1)$ and $\mathbb{I}(s'=s_0)$ represents the indicator function of the event *"the next state is the initial state $s_0$"*. In the literature, this kernel is sometimes referred to as the "restart kernel".
> > >
> > > Next, and we will show that, under this restart kernel, the stationary distribution $\mu(s)$ is **proportional** to the visitation measure $\eta(s)$ (Note: to be more rigorous, we call $\eta(s)$ as a "visitation measure" rather than "visitation distribution", because it's possible that $\eta(s)>1$ for some $s$ and hence not being a proper distribution). The proof is as follows:
> > >
> > > Note that, for $\gamma<1$, the visitation measure at state $s'$ is defined as follows and can be written in a recursive form:
> > >
> > > $$\eta(s'):=\sum_{t=0}^\infty\gamma^t\mathbb{P}\_{\pi}(s_0\rightarrow s',t)=\mathbb{I}(s'=s_0) + \gamma\sum_s\eta(s)\mathbb{P}_{\pi}(s\rightarrow s',1).$$
> > >
> > > It then follows from the definition of $\eta(s)$ that:
> > >
> > > $$\sum_{s'}\eta(s')=\sum_{s'}\sum_{t=0}^\infty\gamma^t\mathbb{P}\_{\pi}(s_0\rightarrow s',t)=\sum_{t=0}^\infty\sum_{s'}\gamma^t\mathbb{P}\_{\pi}(s_0\rightarrow s',t)=\sum_{t=0}^\infty\gamma^t = \frac{1}{1-\gamma}.$$
> > >
> > > Now, we **define** the following distribution $\mu(s)$ by normalizing $\eta(s)$:
> > > $$\mu(s):= \frac{\eta(s)}{\sum_{s'} \eta(s')} = (1-\gamma)\eta(s).$$ Note that $\mu(s)$ is the "proper" visitation distribution.
> > >
> > > In what follows, we will prove that **$\mu(\cdot)$ is indeed the stationary distribution under kernel $\tilde{P}_{\pi}$**. Hence, the visitation measure is *proportional* to the stationary distribution. As a result, the use of stationary distribution in the policy gradient calculation remains valid. To this end, we first note that:
> > >
> > > $$\sum_s\mu(s)\tilde{P}\_{\pi}(s\rightarrow s',1)=\gamma\sum_s\mu(s)\mathbb{P}\_{\pi}(s\rightarrow s',1) + (1-\gamma)\sum_s\mu(s)\mathbb{I}(s'=s_0),$$
> > >
> > > which follows from the definition of $\tilde{P}_{\pi}(s\rightarrow s',1)$. Then, by using the definition $\mu(s):=(1-\gamma)\eta(s)$, we can further re-write the above equation as:
> > >
> > > $$\sum_s\mu(s)\tilde{P}\_{\pi}(s\rightarrow s',1)=(1-\gamma)\left(\gamma\sum_s\eta(s)\mathbb{P}\_{\pi}(s\rightarrow s',1)+\mathbb{I}(s'=s_0)\right)=(1-\gamma)\eta(s')=\mu(s').$$
> > >
> > > This shows that $\mu(\cdot)$ is the **stationary distribution under kernel $\tilde{P}_{\pi}$** and the proof is complete.
> > >
> > > Based on the above insight, we now justify the **correctness** of our revised Lemma 4.2. First according to Sec. 13.2 in [R4], we know that
> > > $$\nabla_\theta J(\theta) = \sum_s \eta(s)\sum_a\nabla\pi_\theta(a|s)Adv_\theta(s,a).$$ Multiplying and dividing the right-hand-side by $1-\gamma$ and using $\mu(s)=(1-\gamma)\eta(s)$, we have that
> > > $$\nabla_\theta J(\theta) = \frac{1}{(1-\gamma)}\sum_s (1-\gamma)\eta(s)\sum_a\nabla\pi_\theta(a|s)Adv_\theta(s,a),$$ which implies that $$\nabla_\theta J(\theta) \propto \sum_s\mu(s)\sum_a\nabla\pi_\theta(a|s)Adv_\theta(s,a).$$ This is exactly what we claimed in the revised Lemma 4.2.
> > >
> > > We thank the reviewer again for these valuable discussions, which strengthens the clarity of our work. We will add the above discussions in the revision to avoid similar doubts.
> > >
> > > ---
> > > **Response to Question 2:** Thanks for your follow-up question. We'd like to point out that $2dm$ comes from the first and the last layer of the DNN. Specifically, the dimension of $H$ for the first layer $x^{(0)}=Hx$ is $dm$, where x is the $d$-dimensional input. Similarly, the dimension of parameter $b$ in the last layer is $m\cdot 1 \le md$. Therefore, the dimension of $\theta^i$ is at most $m(Dm+2d)$.
> > >
> > > ---
> > > [R1] Konda. Actor-critic algorithms. PhD thesis, 2002.
> > >
> > > [R2] Xu et al. Improving sample complexity bounds for (natural) actor-critic algorithms, 2020.
> > >
> > > [R3] Chen et al. Sample and communication-efficient decentralized actor-critic algorithms with finite-time analysis, 2022.
> > >
> > > [R4] Sutton & Barto. Reinforcement learning: An introduction, 1998.

---

### Official Review · Reviewer_wTko · 2025-03-06

**Overall Recommendation:** 3

**Summary:**

This work provides the the first actor-critic algorithm with deep Q-net and deep policy-net for fully decentralized MARL problem, and provides the first global convergence result for such algorithm.

**Claims And Evidence:**

The claim is clear as summarized above, which is supported by theoretical proof (I believe the general proof logic is correct) and experimental results (look comprehensive and convincing).

**Essential References Not Discussed:**

The lits about MARL stops at 2022. Are you sure there are no more after 2022? For example, (Chen et al., 2022) is cited 35 times in Google Scholar. Are some of them related?

**Experimental Designs Or Analyses:**

I briefly scanned the problem settings, hyperparameter values and figures of the experimental results in both the main text and Appendix B. The general process looks clear to me. If possible, it may be better to provide more details, such as the figure of the simulation grid (including the landmark locations), and the math formulation of multi-agent RLHF (see the Question 13 in detail).

**Methods And Evaluation Criteria:**

The experimental settings on simulation and application to RLHF look reasonable and comprehensive. The criterion of reward in the experimental results and the criterion of function value gap in Theorem 5.10 are standard and reasonable.

**Other Comments Or Suggestions:**

(1) The introduction said works including (Chen et al., 2022) only ensures the convergence to some stationary solution. However, (Chen et al., 2022) also obtains the global convergence rate of decentralized natural actor-critic algorithm. You could reword it.

(2) In Theorem 5.10, use $e^{-\Omega(\log^2 m)}$ or $\exp[-\Omega(\log^2 m)]$.

(3) Right after Eq. (8), "we can easily get $-\frac{\langle\nabla J(\theta_t), d_t\rangle}{d_t||} \leqslant-\frac{1}{3}||\nabla J(\theta_t)||+\frac{8}{3}||e_t||$". At the final step of Eq. (8), change $-\frac{8}{3}||e_t||$ to $+\frac{8}{3}||e_t||$. Actually this could be strengthened to
$$-\frac{\langle\nabla J(\theta_t), d_t\rangle}{||d_t||}=-\frac{\langle\nabla J(\theta_t)-d_t, d_t\rangle}{||d_t||}-||d_t||\le ||\nabla J(\theta_t)-d_t||-||d_t-e_t||+||e_t||=-||\nabla J(\theta_t)||+2||e_t||$$ without discussing two cases, but I feel fine if this improvement is not done in the rebuttal.

(4) Right before Eq. (22), "since both $\overline{W}_t$ and $\overline{V}_t$ belong to $\mathcal{B}(B)$".

**Other Strengths And Weaknesses:**

This theoretical result is significant and novel since it is about multi-agent actor-critic algorithms with deep policy network and deep Q-net, which have many practical successes but lack theoretical foundation in existing works. The analysis technique by relating to centralized deep Q-net evaluation (Algorithm 3) is novel. The experiments look comprehensive. The presentation is clear and I can understand well.

The question (1) below in "Questions For Authors" is a major issue which may invalidate Algorithm 2. Also, based on Lemma 5.7, convergence to stationary policy in existing MARL works seems not a limitation as that can imply $\mathcal{O}(\sqrt{\epsilon_{\rm bias}})$-global optimality. Therefore, the global convergence is not novel compared with existing MARL works, especially given that (Chen et al., 2022) actually provides similar global convergence. There also remain some points to be clarified as listed below.

**Questions For Authors:**

(1) (**The major reason for my rejection:**) In Algorithm 2, should the radius $B\le\mathcal{O}(\epsilon)$? If not, the error term (6) of Theorem 5.10 is larger than $B=\Theta(m^{1/32}D^{-6})\ge\mathcal{O}(\epsilon)$. If yes, the weights $W^i(k)$ of all iterates $k$ belong to a very tiny neighborhood $\mathcal{B}(B)$ around the initial parameter $W(0)$, so the output $W^K$ as convex combination of $W^i(k)$ also belongs to this neighborhood. In both cases, it seems that we cannot achieve $\mathcal{O}(\epsilon)$ convergence error even if $\epsilon_{\rm critic}=\epsilon_{\rm actor}=0$. How to explain?

(2) The introduction lists two major technical barriers before the research quesiton, followed by 3 technical challenges after the question. Would the "two major technical barriers" better be "two major limitations of existing works"?

(3) Why are $H$ and $b$ fixed in DNN?

(4) Many other decentralized optimization lits assume that the second largest singular value of the gossip matrix $A$ lies in (0,1). I am curious how does it relate to your Assumption 5.1?

(5) In Assumption 5.3, is there lower bound for reward, e.g., 0 or $-r_{\max}$?

(6) Could you cite at least one paper that provides Lemma 5.4? If there is not such a paper, you could prove it. Also, can we ensure that $\kappa,\rho$ do not rely on the policy?

(7) What's the intuition behind Eq. (14) of Fact A.3? (e.g. from the gradient of the MSPBE objective in Eq. (7))? Does Assumption 5.9 hold for all stationary points $W^*$ satisfying Fact A.3, or there exists one $W^*$ satisfying the Q error bound of $\epsilon_{\rm critic}$?

(8) In Corollary 5.11, is the sample complexity $\Omega(N^2\epsilon^{-3})$?

(9) What is $W_t^i$ in Fact A.2? Does "for the outer loop indexed by $t$" mean "at the $t$-th outer loop of Algorithm 1"?

(10) Should $1^{\top}$ be $\frac{1^{\top}}{N}$ in Eq. (20)?

(11) Is there any theoretical result demonstrating that the total number of DNN parameters can be far less than the total number of state-action pairs?

(12) In the experimental figure results (except Figure 4), does reward mean $\sum_{t=0}^{T-1}\gamma^t r_t$ on the current episode of finite horizon $T$?

(13) In the RLHF experiment, did you use bandit (only 1 time point) or MDP? Is the objective simply the average of policy optimization objective (expected reward with KL penalty) between the two agents? You might provide some math formulations if applicable.

(14) The lits about MARL stops at 2022. Are you sure there are no more after 2022? For example, (Chen et al., 2022) is cited 35 times in Google Scholar. Are some of them related?

**Relation To Broader Scientific Literature:**

This work extends the existing theoretical foundation of actor-critic methods equipped with neural networks from single-agent to multi-agent with new decentralization techniques, and also extends the existing multi-agent actor-critic analysis works from tabular and linear  function approximation to deep Q-network.

**Theoretical Claims:**

I am familiar with existing actor-critic proof analysis. I believe the proof logic is in general similar to that of the existing actor-critic proof analysis and thus correct, though there might be fixable computation errors that I did not find. However, the main issue is in the choice of hyperparameter choice $B$ which may invalidate Algorithm 2, as elaborated in question (1) in "Questions For Authors".

---

> ### Author Rebuttal · Authors · 2025-04-01
>
> Due to the space limitation, we could only respond to a subset of more critical comments in this rebuttal. But we are happy to continue to complete our responses to your remaining comments in the discussion stage when new space opens up.
>
> > **Comment 1:** The introduction said works including (Chen et al., 2022) only ensures…
>
> **Response:** Thanks for pointing out a related work. After reviewing [1], we acknowledge that their dec-NAC approach indeed guarantees global convergence. However, [1] is based on linear function approximation, while we consider nonlinear function approximation. Thus, our work remains the first in the MARL literature that achieves global convergence under **nonlinear function approximation**. We will clarify this in **Sec. 1** (technical barrier) and **Sec. 2** (global convergence) in the revision.
>
> > **Question 2:** Technical typos and Grammar mistakes.
>
> **Response:** Thanks for catching: (1) In Theorem 5.10, we agree that it should be $\exp(-\Omega(\log^2m))$. (2) We will replace "*due to*" with "*since*". (3) Indeed, there is a typo, and the sample complexity should be $\Omega(N^2\epsilon^{-3})$. (4) the correct value is $\frac{1^\top}{N}$. We thank the reviewer’s careful reading and will fix these mistakes in the revision.
>
> > **Question 3:** In Algorithm 2, should the radius $B<\mathcal{O}(\epsilon)$? If not… How to explain?
>
> **Response:** Thanks for your question and we'd like to clarify as follows:
>
> **1)** Rigorously, the reviewer's statement should be written as $B=\mathcal{O}(\epsilon)$, which we fully agree with. However, this does **not** necessarily imply $B\leq\epsilon$, since the Big-O notation hides constant factors.
>
> **2)** We confirm that the updated parameter $W^i$ for each agent $i$ remains within the projection ball centered at $W(0)$ in each critic loop according to Alg.2. However, this does **not** affect the correctness of Theorem 5.10. In our analysis, we are only required to show the estimated value function $\hat{Q}(\cdot;W^i)$ converges to $Q(\cdot;W^*)$, rather than the convergence of $W^i$. This analytical approach is also used by [2].
>
> > **Question 4:** Why are $H$ and $b$ fixed in DNN?
>
> **Response:** To characterize the accuracy of the DNN, we follow the convention in the literature (e.g., [2,4]), which characterized the DNN capability under a fixed $H$ and $b$ setup. In our analysis, it is used for deriving **Eq. 23** in Appendix.A.
>
> > **Question 5:** In Assumption 5.3, is there lower bound for reward?
>
> **Response:** Thanks for catching this. We indeed want to assume $r_t^i \in [0, r_{\max}]$. We will fix this in the revision.
>
> > **Question 6:** Could you cite at least one paper that provides Lemma 5.4…?
>
> **Response:** Thanks for your question. Lemma 5.4 is a mild condition widely used in the literature (e.g., [1,3]). $\kappa,\rho$ are dependent on the policy since they characterize the mixing time of the MDP under the policy.
>
> > **Question 7:** What's the intuition behind Eq. (14) of Fact A.3…?
>
> **Response:** Thanks for your questions and we'd like to further clarify:
>
> 1. **Intuition behind Eq. (14):** The condition in Eq. (14) is analogous to the notion of stationary point in optimization, since $\delta\cdot\nabla_WQ$ serves as the gradient of $W$. Thus, Eq. (14) means that *"there is no descent direction at $W^*$"*.
>
> 2. **Understanding of Assumption 5.9:** This assumption states that for all policy $\theta$, there exists some stationary $W^*$ satisfying the definition in Fact A.3, such that the $\epsilon_{\text{critic}}$ condition holds.
>
> > **Question 8:** What is $W_t^i$ in Fact A.2? Does "for the outer loop indexed by $t$" mean "at the $t$-th outer loop of Algorithm 1"?
>
> **Response:** Thanks for your questions and we'd further clarify. Your understanding is correct: $W_t^i$ is $W^i$ after $K$ iterations in Alg.2 during its $t$-th leveraging in Alg.1. Since Alg.2 is a component of Alg.1, we refer to the $t$-th round of Alg.1 as "*the outer loop indexed by $t$*". Fact A.2 implies that the consensus process does not impact the average value. We will add these clarifications in the revision.
>
> > **Question 9:** Is there any theoretical result demonstrating that the total number of DNN parameters can be far less than the total number of state-action pairs?
>
> **Response:** The answer is yes, which has been shown in [2,3] for single-agent RL. In this work, we generalize this insight to MARL. Please also refer to **Response** to **Question 2 of Reviewer HEYR**.
>
> [1] Chen et al. Sample and communication-efficient decentralized actor-critic algorithms with finite-time analysis, ICML 2022.
>
> [2] Cai et al. Neural temporal-difference and Q-learning provably converge to global optima, NeurIPS 2019.
>
> [3] Gaur et al. Closing the gap: Achieving global convergence (last iterate) of actor-critic under markovian sampling with neural network parametrization, ICML 2024.
>
> [4] Gao et al. Convergence of adversarial training in over-parametrized neural networks, NeurIPS 2019.

---

> > ### Comment · Reviewer_wTko · 2025-04-04
> >
> > I am satisfied with most of your responses except the following:
> >
> > **Question 3 (major reason for rejection):** Since $\widehat{Q}(\cdot; W)$ is an $L$-Lipschitz continuous function of $W$ for a constant $L>0$, $||W^i-W(0)||\le \mathcal{O}(\epsilon)$ implies $||\widehat{Q}(\cdot; W^i)-\widehat{Q}(\cdot; W(0))||\le L\mathcal{O}(\epsilon)$. Hence, if you want $||\widehat{Q}(\cdot; W^i)-Q(\cdot; W^*)||\le L\mathcal{O}(\epsilon)$, we have $||\widehat{Q}(\cdot; W(0))-Q(\cdot; W^*)||\le L\mathcal{O}(\epsilon)+\mathcal{O}(\epsilon)$, which means the initial $W(0)$ should also be $\mathcal{O}(\epsilon)$-optimal, which seems an unrealistic assumption. How to explain?
> >
> > **Question 6:** Would Lemma 5.4 better be an assumption?

---

> > > ### Author Response · Authors · 2025-04-07
> > >
> > > **Response to “major reason”:** Thanks for your question. But it seems the reviewer has some misunderstanding of the Big-O notation $\mathcal{O}(\cdot)$, which represents "growth rate of scaling" rather than some "static value."
> > >
> > > Specifically, let us first recall the **formal definition** of $\mathcal{O}(\cdot)$, which is stated as follows: $f(n)=\mathcal{O}(g(n))$ means that $\exists C>0$ independent of $n$, $\forall n>n_0, |f(n)|\le C g(n)$ [AR1]. Therefore, when talking about $f(\epsilon)=\mathcal{O}(\epsilon)$, we mean that $f(\epsilon)$ is **a member of the family of functions** whose growth rates are upper bounded by $C\epsilon$ for some constant $C>0$, i.e., we are talking about "a function" rather than "a static value." Also, the symbol "$=$" here actually means "$\in$" rather than *"equal."* Also because of this "membership" meaning of the Big-O notation, it is **inaccurate** to write "$f(\epsilon)<\mathcal{O}(\epsilon)$". See the classic textbook [AR1] for further details about the Big-O notation. Also, in the Big-O notation, $C>0$ can be arbitrarily large, as long as it is **independent** of $\epsilon$. So, this implies that $f(\epsilon)=\mathcal{O}(\epsilon)$ **doesn't mean** the value of $f(\epsilon)$ is close to $\epsilon$.
> > >
> > > Therefore, although the reviewer's conclusion $|| W(0)-W^* || = \mathcal{O}(\epsilon)$ is correct, it **doesn't mean** that $W(0)$ is $\epsilon$-distance away from $W^*$. $|| W(0)-W^* || = \mathcal{O}(\epsilon)$ just means that as $\epsilon \rightarrow 0$, $|| W(0)-W^* ||$ will also shrink to $0$ **not slower than a linear fashion** with some slop $C>0$. But please note that the slope $C$ could also be huge.
> > >
> > > [AR1] Thomas H. Cormen, Charles E. Leiserson, Ronald L. Rivest, and Clifford Stein, Introduction to algorithms, 2022.
> > >
> > > ---
> > > **Response to issues about Lemma 5.4:** Thanks for your follow-up question. We acknowledge that in the references we mentioned before, this condition is considered as an assumption, and agree that we can directly assume this condition. However, we'd also like to point out that the existence of stationary distribution is the corollary of irreducible and aperiodic MDP. Therefore, according to Assumption 5.2, we do have Lemma 5.4.
> > >
> > >
> > > ---
> > > **The following Responses are for remaining questions and comments in the first round.**
> > >
> > > > **Question 1:** Does reward mean of finite horizon $T$?
> > >
> > > **Response:** The answer is yes. The reward in those figures are computed using $\sum_{t=0}^{T-1}\gamma^t r_t$ with $T=10$ and $\gamma=0.99$. Here, $T$ has been set with a finite value to reflect the setting of MPE environment.
> > >
> > > > **Question 2:** Many other decentralized optimization lits assume that the second largest singular value of the gossip matrix $A$ lies in (0,1). I am curious how it relates to your Assumption 5.1?
> > >
> > > **Response:** Thanks for your question. While the second-largest eigenvalue condition is commonly used (e.g., [3]), our $\eta$-assumption, derived from [4], is also widely used in the literature (e.g., [6,7]). These two assumptions are alternative conditions and both characterize a well-behaved consensus matrix $A$, ensuring the convergence rate of local values in reaching consensus.
> > >
> > > > **Question 3:** The literature about MARL stops at 2022. Are you sure there are no more...
> > >
> > > **Response:** Thanks for your question. We have checked the papers that cite (Chen et al., 2022). We summarize these works as follows:
> > >
> > > **1)** Aside from those already discussed in our paper, we identified two relevant works [7,8], both of which employ linear approximation and neither established any global convergence result.
> > >
> > > **2)** Refs. [1-6,9,10] primarily focused on  federated RL, robust decentralized RL, and multi-agent Markov games, which are not directly related to our work.
> > >
> > > We will cite these works in the revision.
> > >
> > > [1] Neural temporal-difference and Q-learning provably converge to global optima.
> > >
> > > [2] Convergence of adversarial training in over-parameterized neural networks.
> > >
> > > [3] Randomized gossip algorithms.
> > >
> > > [4] Distributed subgradient methods for multi-agent optimization.
> > >
> > > [5] Improving sample complexity bounds for (natural) actor-critic algorithms.
> > >
> > > [6] Finite-Time Convergence and Sample Complexity of Multi-Agent Actor-Critic Reinforcement Learning with Average Reward.
> > >
> > > [7] Sample and communication efficient fully decentralized marl policy evaluation via a new approach: Local td update.
> > >
> > > [8] Learning to coordinate in multi-agent systems: A coordinated actor-critic algorithm and finite-time guarantees.
> > >
> > > [9] Closing the gap: Achieving global convergence (last iterate) of actor-critic under markovian sampling with neural network parametrization.
> > >
> > > [10] Sample and communication-efficient decentralized actor-critic algorithms with finite-time analysis.

---

### Official Review · Reviewer_cwpA · 2025-03-14

**Overall Recommendation:** 2

**Summary:**

Summary:
Goal of the paper is to develop a decentralized MARL (dec-MARL) Actor-critic (AC) algorithm, with DNN critic that achieves global optimality

Technical challenges to developing dec-MARL:
- AC methods from single-agent RL are inadequate for MARL due to distributed nature
- Even if first challenge was alleviated, the compounding of error from non-linear estimators is big resulting in inaccurate method
- Gradients driven from descent lemma are not sufficient to achieve global convergence

Paper contributions:
- Development of an AC based dec-MARL algorithm that achieves a global optimality convergence rate of O(1/T).
- The algorithm uses the following components: DNN critic that is trained using TD learning, actor trained using policy gradient based method, a gossiping technique for decentralized agents to communicate.
- The paper also presents some empirical experiments to show performance of the proposed algorithm.

**Claims And Evidence:**

The paper claims the following:
- The first AC-based decentralized algorithm for MARL with DNN (non-linear function approximation) based critic
- Unlike past literature, they achieve global convergence

Evidence:
- They provide proof of theorem for the rate of achieving global convergence
- They also provide ablation study and experiments to show the algorithm's performance

**Essential References Not Discussed:**

See above comment on MAPPO, MATD3 etc.

It would be great to discuss the differences with these in related works.

**Experimental Designs Or Analyses:**

I didn’t check in detail the validity of experimental designs or theoretical analysis.


One comment on the analysis is, the authors argue that based on the results increasing the depth of the DNN improves performance, however, that is not the case in Figure. 7.

**Methods And Evaluation Criteria:**

The paper introduced experiments on two Benchmarks, one simple MPE environment (Simple Spread) and an LLM-based task. Both are reasonable benchmarks and the first is used extensively in literature.


However, the evaluation criteria are missing a comparison of the method to existing MARL algorithms to show performance differences.

**Other Comments Or Suggestions:**

I didn’t find typos.

**Other Strengths And Weaknesses:**

## Strengths:
- Theoretical analysis of the presented algorithm which is lacked in MARL literature
- The use and analysis of non-linearity in critic is relevant ro MARL applications due to large state and action spaces.
- The paper presented some interesting insights for practical use

## Weaknesses:
- Abstract unclear for which setting the global convergence is for
- Clarify the  “non-competing” nature of the rewards, otherwise, the statements are wrong
- Paper flow in introducing contributed assumptions and lemmas is not clear (Ex. Lemma 7 is not clear if it is from the referenced paper or a contribution of this work)

### Main concern:

Unless I misunderstood, agents learn independently from others, but the algorithm uses all agents’ actions. Hence, the setting does not seem fully decentralized.  In particular, in lines 6 and 7 from Alg. 2, the critic has access to global actions which are commonly used in algorithms that are usually centralized.
This affects the relevance of the considered problem, please see my question below.

**Questions For Authors:**

1. How do your obtained results compare to the state of the art of 1-3 relevant settings, such as decentralized minimization, or decentralized min-max games?
2. What were the main challenges in extending the XXX (convergence and complexity) results from the above settings?
3. (related to the main concern) Could you give examples of applications that satisfy the considered setting where the critic has access to all agents’ actions, but agents (nodes) operate independently?
4. What are the benefits of using a decentralized algorithm over Centralized training decentralized execution algorithms, and more broadly what is the motivation behind proposing a new decentralized algorithm?

**Relation To Broader Scientific Literature:**

I am aware that several actor-critic-based algorithms, such as MAPPO, MATD3, and COMA, have been previously introduced for MARL and are widely used in practice.
Many of the introduced algorithms for MARL in the past lack theoretical foundations which is the focus of this paper.

The paper also mentions improvements over past works in dec-MARL:
- past only dealt with linear function approximation for critics while their algorithm deals with a broader class of non-linear functions.
- Their convergence rate is still reasonable despite the non-linearity and global optimality guarantee
- They discuss some findings for practical use from the analysis such as: improvement from the use of TD over Q-learning and effect of depth on DNN on performance. Which provided different results than past literature

**Theoretical Claims:**

The theoretical claims appear to be reasonable, although I did not verify the proofs. Additionally, as mentioned in my comment 3 below, the structure and presentation of the paper make it unclear which contributions are from previous work and which are from this work.

---

> ### Author Rebuttal · Authors · 2025-04-01
>
> We thank the reviewer for the constructive comments! We address the reviewer's comments point by point as follows:
>
> > **Comment 1:** The evaluation criteria are missing a comparison of the method to existing MARL algorithms to show performance differences. It would be great to discuss the differences with these in related works (such as MAPPO, MATD3, and COMA).
>
> **Response:** The algorithms mentioned by the reviewer all follow a centralized training decentralized execution (CTDE). By contrast, our approach is based on consensus-based **decentralized** training. Since our work executes learning without any centralized task, it fundamentally differs from these related works and hence is not directly comparable.
>
> > **Comment 2:** The structure and presentation of the paper make it unclear which contributions are from previous work and which are from this work (such as Lemma 5.7).
>
> **Response:** We would like to clarify that Lemma 5.7 is directly cited from a prior work. We have explicitly stated, “*These two assumptions provide the following key lemma (Agarwal et al., 2021).*” To further improve clarity, we will underline all citations in the next revision.
>
> > **Comment 3:** The authors argue that based on the results increasing the depth of the DNN improves performance, however, that is not the case in Figure 7.
>
> **Response:** We confirm that our theorem indeed suggests that increasing the depth of the DNN benefits performance. Although it may not be easy to see this from Fig. 7, the overall trend aligns with our theoretical predictions: deeper networks $D$ generally perform better. To further explain this phenomenon, we note that certain theoretical conditions may not strictly hold in numerical experiments, which could account for the inconspicuous gap between our theoretical results and empirical observations.
>
> > **Comment 4:** Abstract unclear for which setting the global convergence is for. Besides, clarify the “non-competing” nature of the rewards, otherwise, the statements are wrong.
>
> **Response:** We confirm that our work considers the **cooperative** setting, where multiple agents collaborate to maximize the long-term global cumulative reward. We acknowledge that our convergence result is specifically derived for this non-competing scenario. We thank the reviewer for pointing out this and will clarify this in **Abstract** in our revision.
>
> > **Question 5:** Agents learn independently, but the algorithm uses all agents’ actions. In particular, the critic has access to global actions. This affects the relevance of the considered problem. Could you give examples of applications that satisfy the considered setting where the critic has access to all agents’ actions, but agents operate independently?
>
> **Response:** One example can be found in the carrier sensing multiple access (CSMA) protocol for wireless random access networks, where participating devices (agents) can listen to packet signals transmitted over the channel and be aware of the actions other devices have taken. In general, applications may include any system where the agents can observe one another (this is also needed for consensus) for information sharing, e.g., robotics, drones, vehicles, etc. In addition, we’d like to point out that joint action is widely assumed in the MARL literature [1-3].
>
> [1] Zhang et al. Fully decentralized multi- agent reinforcement learning with networked agents, ICML 2018.
> [2] Zeng et al. Learning to coordinate in multi-agent systems: A coordinated actor-critic algorithm and finite-time guarantees. In Learning for Dynamics and Control Conference, 2022.
> [3] Wai et al. Multi-agent reinforcement learning via double averaging primal-dual optimization, NeurIPS 2018.
>
> > **Question 6:** How do your obtained results compare to the state of the art of 1-3 relevant settings? Besides, What were the main challenges in extending the results from the above settings?
>
> **Response:** We appreciate the reviewer’s suggestions. While these are indeed interesting new directions for future work, establishing a global optimality theory for these new decentralized optimization problems is highly non-trivial and deserves dedicated papers. To provide some high-level intuition, we believe that the new handling of nonlinear approximation and consensus techniques: **(1)** will be important to the analysis for reasons similar to those illustrated in our Fig. 1; **(2)** can potentially be combined with other techniques, such as momentum, to further enhance performance.
>
> > **Question 7:** What are the benefits of using a decentralized algorithm over CTDE algorithms, and more broadly what is the motivation behind proposing a new decentralized algorithm?
>
> **Response:** CTDE algorithms require that there must exist an entity for executing centralized tasks. Such a requirement may render the algorithms only applicable to a limited number of real-world scenarios. A fully decentralized algorithm on the other hand does not require a central entity.

---

### Decision · Program_Chairs · 2025-05-01

**Decision:**

Accept (poster)

**Comment:**

This paper presents theoretical results showing convergence to a globally optimal solution for fully observable MARL at a rate of O(1/T). Experiments are also provided showing the performance of methods.

The theoretical results in this paper are strong. To the best of my knowledge, this is the first paper to show global optimality using deep learning in MARL. There are strong assumptions to allow this result to happen though and these should be made more clear early in the paper (e.g., in the introduction). For example, the setting is fully observable, a non-standard multi-agent MDP structure is used, and there is communication between the agents during training.  This last one is the most important to clarify since is easily can get lost and is crucial to the result. Because of this assumption, this method would typically be considered centralized training for decentralized execution rather than purely decentralized MARL. The authors call this consensus-based decentralized training in the response but this is not a standard paradigm and not explained in the paper. All of this should be clarified in the paper.

There were several other issues raised by the reviews such as a more detailed discussion of related work and details of the proofs. The author response was helpful in this regard but the paper should also be clarified.